# Sex-specific regulatory architecture of pancreatic islets from subjects with and without type 2 diabetes

Mirza Muhammad Fahd Qadir [1,2,3], Ruth M Elgamal [4,5], Kejing Song[6], Parul Kudtarkar[5], Siva S V P Sakamuri [7], Prasad V Katakam[7], Samir S El-Dahr [8], Jay K Kolls[6], Kyle J Gaulton[5] & Franck Mauvais-Jarvis [1,2,3]✉

## Abstract

Patients with type 2 and type 1 diabetes (T2D and T1D) exhibit sex-specific differences in insulin secretion, the mechanisms of which are unknown. We examined sex differences in human pancreatic islets from 52 donors with and without T2D combining single cell RNA-sequencing (scRNA-seq) and single nucleus ATAC-sequencing (snATAC-seq) with assays probing hormone secretion and bioenergetics. In non-diabetic (ND) donors, sex differences in islet cell chromatin accessibility and gene expression predominantly involved sex chromosomes. In contrast, islets from T2D donors exhibited similar sex differences in sex chromosome-encoded differentially expressed genes (DEGs) as ND donors, but also exhibited sex differences in autosomal genes. Comparing β cells from T2D and ND donors, gene enrichment of female β cells showed suppression in mitochondrial respiration, while male β cells exhibited suppressed insulin secretion, suggesting a role for mitochondrial failure in females in the transition to T2D. We finally performed cell type-specific, sex stratified, GWAS restricted to differentially accessible chromatin peaks across T2D, fasting glucose, and fasting insulin traits. We identified that differentially accessible regions overlap with T2D-associated variants in a sex- and cell type-specific manner.

**Keywords** Diabetes; Islet; OMICS; Sex Differences; Precision Medicine
**Subject Categories** Chromatin, Transcription & Genomics; Metabolism; Molecular Biology of Disease

## Introduction

Type 1 and type 2 diabetes (T1D, T2D) are heterogeneous diseases, and biological sex affects their pathogenesis. In the context of T2D, sex affects the development of adiposity, insulin resistance, and dysfunction of insulin-producing β cells of pancreatic islets (Gannon et al, 2018; Mauvais-Jarvis, 2015). For example, ketosis-prone diabetes is a form of T2D with acute β cell failure and severe insulin deficiency predominantly observed in black men (Louet et al, 2008; Mauvais-Jarvis et al, 2004; Umpierrez et al, 2006). A missense mutation in the β cell-enriched MAFA transcription factor is found in subjects with adult-onset β cell dysfunction, where men are more prone to β cell failure than women (Walker et al, 2021). Similarly, T1D is the only common autoimmune disease characterized by a male predominance (Mauvais-Jarvis, 2015, 2017, 2018), and males who develop T1D during puberty have lower residual β cell function than females at diagnosis (Samuelsson et al, 2013). Furthermore, among T1D subjects receiving pancreatic islet transplantation, recipients of male islets exhibit early graft β cell failure when compared to recipients of female islets (Lemos et al, 2022). The mechanisms that drive preferential β cell failure in males, however, is unknown. Studying sex differences in islet biology and dysfunction represent a unique avenue to understand sex-specific heterogeneity in β cell failure in diabetes (Gannon et al, 2018).

Female- and male-specific blood concentrations of the gonadal hormones estradiol and testosterone produce differences in islet function in vivo (Mauvais-Jarvis, 2011; Navarro et al, 2016; Navarro et al, 2018; Tiano et al, 2011; Tiano and Mauvais-Jarvis, 2012; Wong et al, 2010; Xu et al, 2018; Xu et al, 2019; Xu et al, 2023b). However, the sex-specific and cell autonomous factors that influence islet function outside the in vivo hormonal environment are unknown. These differences could be due to sex chromosome gene dosage, or epigenetic programming caused by testicular testosterone during development in males (Mauvais-Jarvis, 2015; Mauvais-Jarvis et al, 2020; Mauvais-Jarvis et al, 2021). The Genotype-Tissue Expression (GTEx) project analysis of the human transcriptome across various tissues revealed that the strongest sex bias is observed for X-chromosome genes showing higher expression in females (Oliva et al, 2020). In the pancreas, the majority of genes with sex-biased expression are on the sex chromosomes and most sex-biased autosomal genes are not under direct influence of sex hormones (Mayne et al, 2016). In human

[1]Section of Endocrinology and Metabolism, John W. Deming Department of Medicine, Tulane University School of Medicine, New Orleans, LA, USA. [2]Southeast Louisiana Veterans Health Care System, New Orleans, LA, USA. [3]Tulane Center of Excellence in Sex-Based Precision Medicine, New Orleans, LA, USA. [4]Biomedical Sciences Graduate Program, University of California, San Diego, La Jolla, CA, USA. [5]Department of Pediatrics, University of California, San Diego, La Jolla, CA, USA. [6]Center for Translational Research in Infection and Inflammation, John W. Deming Department of Medicine, Tulane University School of Medicine, New Orleans, LA, USA. [7]Department of Pharmacology, Tulane University School of Medicine, New Orleans, LA, USA. [8]Department of Pediatrics, Tulane University, School of Medicine, New Orleans, LA, USA. ✉E-mail: fmauvais@tulane.edu

pancreatic islets, DNA methylation of the X-chromosome is higher in female than males (Hall et al, 2014). Thus, the cell autonomous influence of sex chromosome genes may impact sex-specific islet biology and dysfunction and diabetes pathogenesis.

Here, we examined sex and race differences in human pancreatic islets from up to 52 donors with and without T2D using an orthogonal series of experiments including single-cell RNA-seq (scRNA-seq), single nucleus assay for transposase-accessible chromatin sequencing (snATAC-seq), and dynamic hormone secretion and bioenergetics. Our studies establish biological sex as a genetic modifier to consider when designing experiments of islet biology.

## Results

### Human islet cells show conserved autosomal gene expression signatures independent of sex and race

We performed scRNA-seq on pancreatic islets from age- and BMI-matched non-diabetic donors across race/ethnicity and sex (Tulane University Islet Dataset, TUID, $n = 15$), which we combined with age- and BMI-matched non-diabetic donors and donors with T2D from the HPAP database (Elgamal et al, 2023; Kaestner et al, 2019) ($n = 37$) to create an integrated atlas of islet cells (Fig. 1A; Appendix Fig. 1A,B). To obtain high-quality single cell signatures, we used a series of thresholds including filtering, ambient RNA correction, and doublet removal, resulting in 141,739 high-quality single cell transcriptomes, with TUID showing equal or greater sequencing metrics than HPAP (Appendix Fig. 1C,D). We identified 17 cell clusters, which we annotated based on marker genes with differential expression (DEGs) correlating to known transcriptional signatures of islet cells (Fig. 1B) (Van Gurp et al, 2022). Cell clusters showed even distribution across sex, race/ethnicity, disease, and library of origin (Fig. 1C). Consistent with a prior analysis (Elgamal et al, 2023), all islet cell clusters except for lymphocytes and Schwann cells were identified in HPAP data (Appendix Fig. 1B). Notably, we observed greater variability in total cell number within each donor library in HPAP compared to TUID (Fig. 1D). We observed a high degree of correlation between cell-specific gene expression and cell clusters across donors (Appendix Fig. 1E). As expected, sex chromosome-specific transcripts were expressed across male and female cell types (Appendix Fig. 1F).

We more broadly examined DEGs across clusters by creating sample 'pseudo-bulk' profiles for each cell type. The gene counts of each specific cell type per donor were aggregated into one gene count profile. This enabled us to control for the disproportionate cell numbers and related gene counts across donors. This also avoided the pseudo-replication of cells and their genes being repetitively sampled from a fixed donor (Fig. 1D). The expression of autosomal genes specific to each cell cluster was independent of sex and race/ethnicity. Endocrine cells exhibited consistent cell type-specific markers expression independent of sex for the ten most upregulated genes, β (*INS*, *MAFA*), α (*GCG*, *ARX*), δ (*HHEX*, *SST*), ε (*GHRL*), γ (*PPY*), cycling endocrine (*TOP2A*, *MKI67*) (Fig. 1E; Appendix Fig. 1G). Similarly, non-endocrine cell types, exhibited consistent cell type-specific upregulated genes independent of sex for ductal (*CFTR*, *TFF1*), acinar (*PNLIP*, *AMY2A*), quiescent stellate (*PTGDS*, *DCN*), activated stellate (*RGS5*, *FABP4*), endothelial (*PECAM1*, *VWF*), lymphocyte

(*CCL5*, *CD7*), macrophage (*SDS*, *FCER1G*), mast (*TPSB2*, *TPSAB1*), and Schwann cells (*SOX10*, *CDH19*) (FDR < 0.1) (Fig. 1E; Appendix Fig. 1G). Using cell type-specific DEGs, we identified upregulated cell type-specific pathways using the gene ontology database (FDR < 0.2) (Carlson et al, 2019). Endocrine cells were enriched in peptide hormone secretion independent of sex and race/ethnicity (Fig. 1F). Other cell types showing upregulated cell type-specific pathways included cycling endocrine cells (mitotic cell cycle transition, organelle fission), ductal cells (organic anion transport, branching morphogenesis), acinar cells (digestion, alcohol metabolism), quiescent stellate cells (collagen fibril organization, muscle cell differentiation), activated stellate cells (cell proliferation, cell chemotaxis), endothelial cells (endothelial cell migration, angiogenesis), lymphocytes (immune receptor signaling, T-cell selection), macrophages (antigen processing and presentation, cell chemotaxis), mast cells (immune response, mast cell activation) and Schwann cells (CNS myelination and axon development) (Fig. 1F). Cell network analysis confirmed segregation of endocrine pathways from exocrine and immune cell type pathways (Appendix Fig. 1H). Taken together our data demonstrate that canonical gene networks are conserved across endocrine and non-endocrine cell types independent of sex and race/ethnicity (Fig. 1F; Appendix Fig. 1H).

### Sex differences in islet cell transcriptomes from non-diabetic donors predominantly affect sex chromosome genes

We performed two sets of analysis comparing changes in gene expression in biological variables of sex and race across groups. To study transcriptional differences across donors, we generated principal component analysis (PCA) plots of islet 'pseudo-bulk' transcriptional profiles across all 52 donors. Donors did not cluster based on sex, race/ethnicity, disease status, or origin of donor (Fig. 2A). We next segregated donors by cell type, and the resulting PCA showed clustering of samples based on cell type (Fig. 2B). Both whole islet 'pseudo-bulk' and individual cell type 'pseudo-bulk' sample profiles showed no clustering based on sex or race. This suggests that human islets do not exhibit major sex or race/ethnicity differences in cell type transcriptional profiles.

Focusing on non-diabetic donors, we examined genes with differences in expression between sexes using cell type 'pseudo-bulk' analysis. Most sex-associated genes were related to sex chromosomes (FDR < 0.1). In β cells, 60% of genes with increased expression in females were linked to the X chromosome and 70% of genes increased in males were linked to the Y chromosome (Fig. 2C; Appendix Fig. 2A). Similarly, in α cells 50% of male- and 57% of female-enriched genes were linked to the X or Y chromosome, respectively (Fig. 2D; Appendix Fig. 2A). In α/β cells, X-inactive specific transcript (*XIST*) and lysine demethylase 6A (*KDM6A*) were upregulated in females, while ribosomal protein S4 Y-linked 1 (*RPS4Y1*) and lysine demethylase 5C (*KDM5D*) was upregulated in males (Fig. 2C,D). The few sex-associated autosomal genes in α/β cells exhibited a small effect size compared to sex-associated X and Y genes (Fig. 2C,D). We only observed significant race/ethnicity differences in DEGs between hispanic and white β and α cells (Appendix Fig. 2C).

Next, we identified sex-specific changes in pathways related to sex chromosome genes using gene set enrichment analyses (Fig. 2E; Appendix Fig. 2B). Female β cells were enriched for pathways for

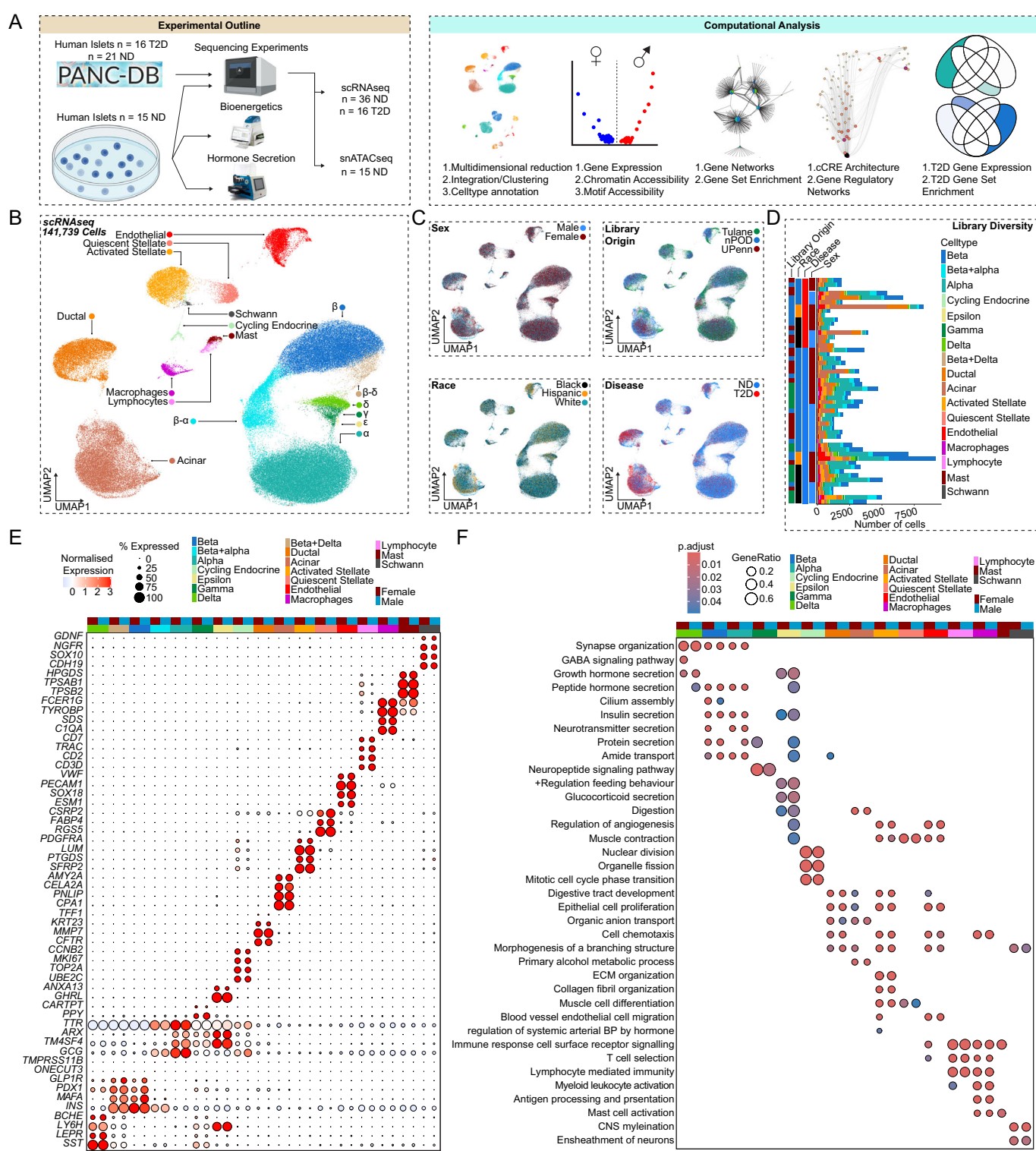

X-chromosome inactivation and histone lysine demethylation, whereas male β cells were enriched for pathways for Y-chromosome genes, histone lysine demethylation, and male sex determination (Fig. 2E). Female α cells were enriched for histone lysine demethylation, X-chromosome inactivation, and mitochondrial transcription, while male α cells were enriched for histone

demethylase activity (Fig. 2F). Similar effects were observed in other cell types (Appendix Fig. 2B). Race differences in islet cells are shown in Fig. 2E,F as well as Appendix Fig. 2C,D. Of note, black male β cells showed higher cytokine signaling compared to white males, suggesting black male β cells may exhibit a higher inflammatory response (Fig. 2E).

**Figure 1. Pancreatic islet cells have a conserved expression signature across sex and race.**

(A) Experimental and computational design. (B) UMAP plot denoting integrated clustering of 141,739 single pancreatic islet cells across 17 clustered cell types based on their scRNAseq profiles, spanning n = 52 datasets. Each cluster cell type is denoted by a label and color. (C) Cells diversified based on donor's sex, origin, race, and disease status. (D) Cell number stemming from each of the n = 52 donors, grouped based on origin, race, disease status, and sex. (E) Gene expression dotplot of select top differentially expressed genes amongst non-diabetic donors across sex. Dotplot is clustered based on disease, source, sex, and race, as denoted by the bars on top. Select genes are labeled on the y-axis. (F) Gene ontology (GO) analysis showing select upregulated pathways across clusters and sex as shown in (E). The intensity of the color denotes scaled FDR corrected adj p-value, and size of the bubble denotes the gene:query ratio. Non-diabetic females: 20, non-diabetic males: 16, T2D females: 9, T2D males: 7. DEGs have FDR adjusted q-value < 0.1, GO pathways have FDR adjusted q-value < 0.2. A Hypergeometric statistical test (clusterProfiler) used for over representation analysis in (F). All replicates are biological in F1 (n = 52, non-diabetic females: 20, males: 16, T2D females: 9, T2D males: 7).

## Accessible chromatin landscape across islet cells

To examine the effect of sex on the epigenome, we performed snATAC-seq on all non-diabetic donors of the TUID. To confirm library quality, we filtered and evaluated single nuclei across all 15 donors for TSS enrichment, fragment of reads in promoters, and fragment reads in accessible peaks (Appendix Fig. 3A,B), as well as sample specific sequencing metrics (Appendix Fig. 3C,D). We then clustered the 52,613 filtered profiles resulting in 11 distinct cell clusters which, like gene expression data, were evenly distributed across sex, race/ethnicity, and donor (Fig. 3A–C). To determine the identity of each cluster, we used label transfer to annotate each snATAC-seq cell cluster using our integrated scRNAseq islet cell atlas as a reference. We observed a high degree of correlation between genes with differential accessibility in snATAC-seq and genes with differential expression scRNAseq (Fig. 3D). Cell types also showed a high degree of correlation between RNA expression, chromatin accessibility, and predicted RNA expression (Appendix Fig. 3E–G). We further examined the cell type annotations using the activity of cell type-specific genes. This validated clusters representing β (INS-IGF2), α (GCG), δ (SST), γ (PPY), acinar ductal (CFTR), (PRSS1), endothelial (ESM1), macrophage (SDS), stellate PDGFRA), and lymphocyte (CD3D) cells by comparing gene accessibility with predicted RNA expression (Fig. 3E,F; Appendix Fig. 3H).

To characterize regulatory programs across each cluster, we identified candidate cis-regulatory elements (cCREs) in each cell type resulting in 404,697 total cCREs across all 11 cell types. We next identified cCREs with activity specific to each cell type, resulting in 55,710 cell type-specific cCREs (Fig. 3G). We identified genes in proximity to cell type-specific cCREs, resulting in a list of putative gene targets of cell type-specific regulatory programs. Evaluating these gene sets for enrichment of gene ontology terms revealed cell type-specific processes, which were similar to those identified in cell type-specific gene expression (Fig. 3H). Using chromVAR (Schep et al, 2017), we identified transcription factor (TF) motifs enriched in the accessible chromatin profiles of each cell type using the JASPAR 2020 database (Fornes et al, 2020). In-depth analysis of these motifs revealed cell type-specific TF motif enrichment patterns (Fig. 3I). For example, we observed enriched motifs for ISL1 in endocrine cells, PDX1 in β and δ cells, and SOX9 in ductal and acinar cells (Fig. 3I,J). These accessible motifs also paralleled cell type-specific TF expression in scRNA-seq (Fig. 3J). Similar to previous studies (Balboa et al, 2022; Chiou et al, 2021; Muraro et al, 2016; Segerstolpe et al, 2016), hierarchical motif clustering highlighted that the regulatory programs of β and δ cells are more related, as with α and γ cells (Fig. 3G). Select motifs highly enriched for a cell type (fold enrichment > 1.5, −log10 FDR > 50) included PAX4, RFX2, NKX6-2, and PDX1 in β cells, NKX6-2,

NKX6-1, PDX1, and MEOX1 in δ cells, MAFB, FOXD2, and GATA2-5 in α cells, and KLF15 and NRF1 in γ cells (Appendix Fig. 3I). Non-endocrine cells motif enrichments are also provided in Appendix Fig. 3I.

## Sex differences in chromatin accessibility of islet cells from non-diabetic donors predominantly affects sex chromosomes

To assess sex differences in chromatin accessibility, we identified sex-associated cCREs using logistic regression. As expected, β cells exhibited sex differences in chromatin accessibility at sex chromosome genes including KDM6A, XIST, and KDM5D (Fig. 4A). Males exhibited more differentially accessible regions (250 in β, 565 in α) than females (203 in β, 553 in α). Next, we identified genes in a 100 kb proximity to sex-associated cCREs and interrogated their RNA expression. We found that Y-linked genes (SRY, RPS4Y1, UTY, TTTT14) in males and X-linked genes (KDM6A, XIST, DHRSX) in females were proximal to sex-associated cCREs (Fig. 4B). Accordingly, when comparing gene expression and cCREs with sex-specific association, we predominantly observed sex-chromosome genes (Fig. 4C). Gene ontology analysis of this subset of genes revealed enrichment in pathways regulating epigenetic control and X chromosome dosage compensation in females, and histone modification in males (Fig. 4D). Notably, the histone demethylase X-linked gene KDM6A and the long non-coding RNA XIST were more accessible in female islet cells, while the histone demethylase Y-linked gene KDM5D was more accessible in males (Fig. 4E). We examined sex differences in TF-specific motif accessibility in α/β cells. Notably, females exhibited a greater number of TF-specific accessible motifs (511 in β, 376 in α) compared to males (33 in β, 74 in α) (Fig. 4F). Upon interrogating differentially expressed TF across cell types, MAFA, SIX3, PDX1, and RXRG were upregulated in β cells while ARX, FEV, STAT4, and ISL1 were upregulated in α cells irrespective of sex (Fig. 4G). We applied Pando (Fleck et al, 2023) to scRNA-seq and snATAC-seq data to infer relationships between target gene expression, TF activation, and TF binding and define gene regulatory networks (GRNs) in male and female β and α cells. The GRNs provide sets of regulated target genes and cCREs for expressed TFs. Irrespective of sex, MAFA, BHLHE41, MEIS2, and MLXIPL in β cells, and PAX6 and SOX5 in α cells, exhibited a high degree of centrality and revealing many associated genes within these TF GRNs (Fig. 4H). In males, PDX1, NKX6-1, and RXRG exhibited higher centrality in β cells, and ARX exhibited higher centrality in α cells, compared to females (Fig. 4H).

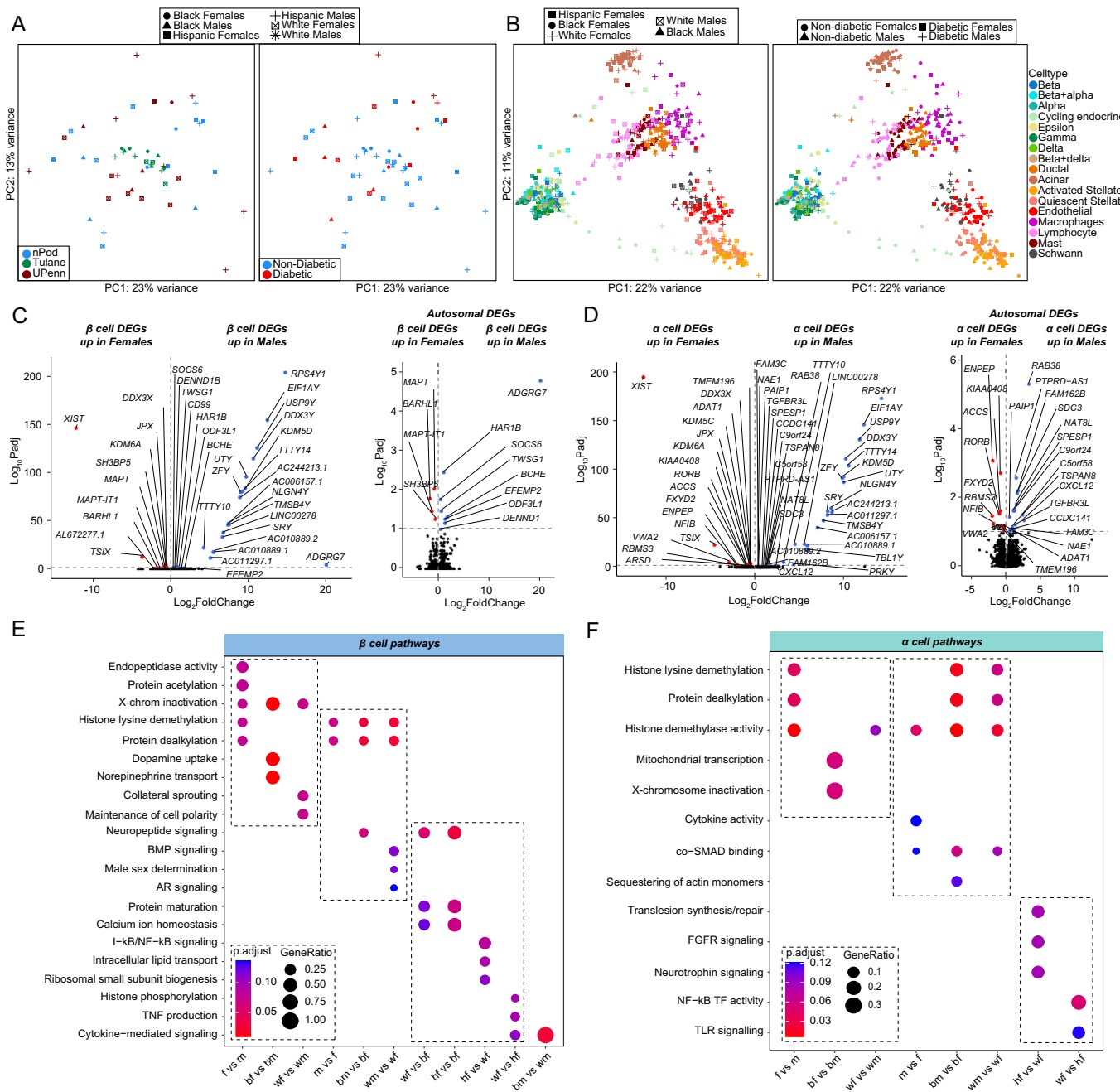

**Figure 2. Transcriptional differences across islet β and α cells, highlight enrichment in sex-chromosome genes.**

(A) Principal component analysis (PCA) plot of pseudo-bulk transcriptional profiles across all individual donor islets. (B) PCA plot of pseudo-bulk transcriptional profiles in each cell type across all donors. (C, D) Volcano plots showing all differentially expressed genes (DEGs) (left panel) or autosomal DEG subset (right panel) across sex in non-diabetic: (C) β cells. (D) α cells. (E) GO analysis of all β cell DEGs. (F) GO analysis of all α cell DEGs. DEGs have FDR adjusted q-value < 0.1, GO pathways have false discovery rate (FDR) adjusted q-value < 0.2. A Wald statistical test (DESeq2) was used for (C, D) with a Benjamini–Hochberg post hoc correction (FDR). A Hypergeometric statistical test (clusterProfiler) used for over representation analysis in (E, F). All replicates are biological in (C, D) (n = 52, non-diabetic females: 20, males: 16, T2D females: 9, T2D males: 7).

## Sex and race differences in β cell function

We performed dynamic insulin and glucagon secretion assays in TUID islets for non-diabetic donors. We observed a decreased insulin response to high glucose and IBMX (a phosphodiesterase inhibitor which raises intracellular cAMP) in black male compared

to white male islets (Fig. 5A,B). There was no significant difference in insulin secretion across sex and race using other classical insulin secretagogues (Fig. 5A–D) or an ascending glucose concentration ramp (Appendix Fig. 4A–D). We observed no difference of race or sex on α cell function using classical glucagon secretagogues, although females exhibited a trend toward higher glucagon

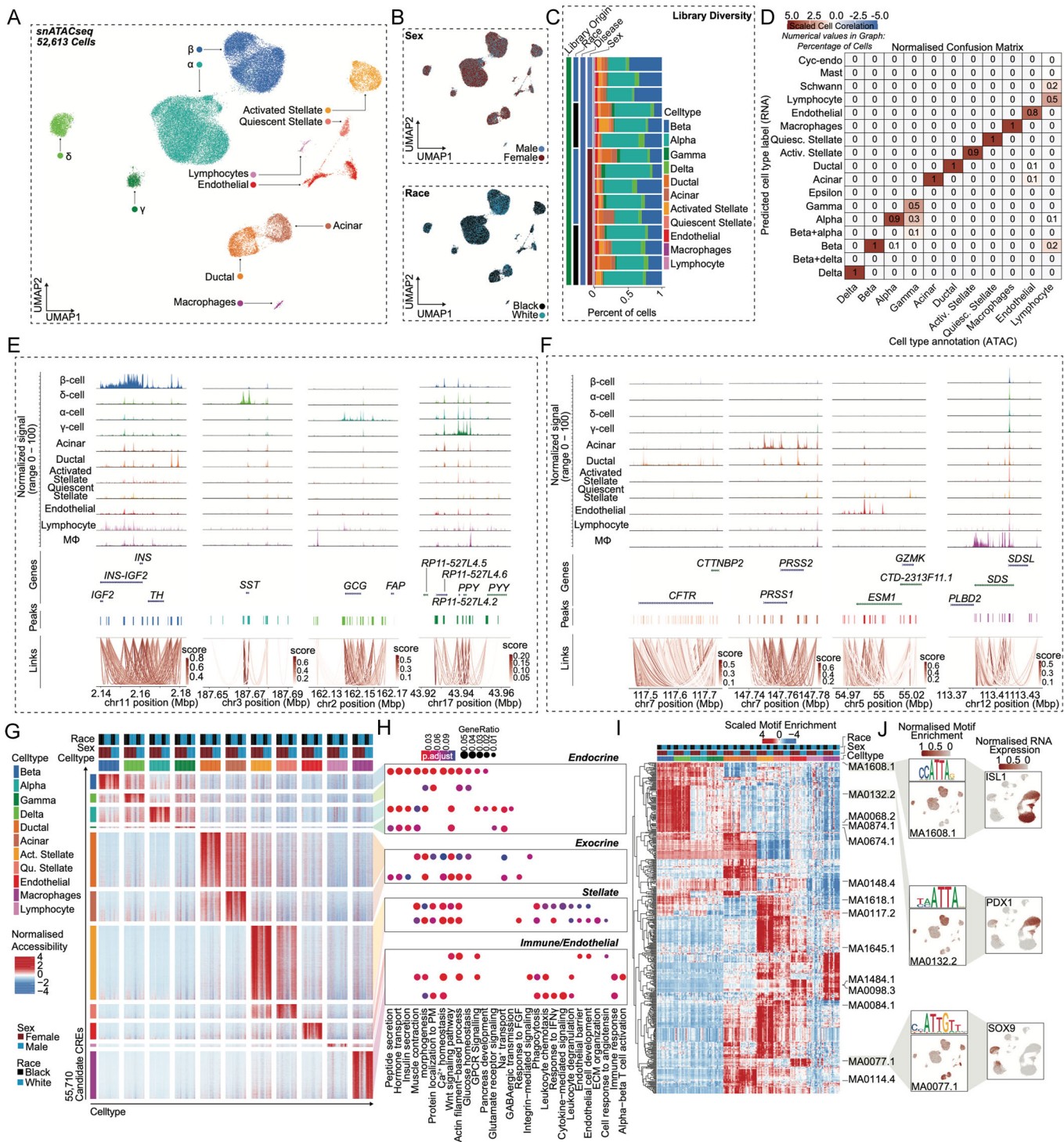

secretion (Fig. 5E–H). We also examined the effects of sex and race on islet bioenergetics by quantifying oxygen consumption rate (OCR) and extracellular acidification rate (ECAR) during a glucose challenge in TUID islets. Female islets exhibited greater ATP-mediated respiration and coupling efficiency than male islets (Fig. 5I–L), suggesting more efficient mitochondria. There was no difference in ECAR between male and female islets (Appendix Fig. 4E–H).

## Dysregulation of β and α cell transcriptomes from non-diabetic compared with T2D donors suggests sex differences in T2D pathogenesis

We examined the effect of sex on islet hormone secretion using the HPAP islet perifusion database matched for donors we sequenced in this study. Islets from male and female donors with T2D exhibited decreased insulin secretion in response to high glucose,

**Figure 3. Chromatin accessibility landscape of human pancreatic islet cell types.**

(A) UMAP plot denoting integrated clustering of 52,613 single pancreatic islet cells across 11 clustered cell types based on their accessible chromatin profiles, spanning $n = 15$ datasets. Each cluster cell type is denoted by a label and color. (B) Cell diversified based on sex and race. (C) Cell distribution stemming from each of the $n = 15$ donors, grouped based on race and sex. (D) Normalized confusion matrix, showing correlation across cell types based on their cell annotation based on their accessible chromatin profile (x-axis) and predicted cell type label gene expression profile (y-axis). (E) Chromatin accessibility density profile (aggregated peak reads for each cell type) within a 50-kb window flanking a transcriptional start site (TSS) for selected endocrine marker genes. (F) Promoter accessibility as in (E) for selected acinar, ductal, endothelial, and macrophage genes. (G) Row normalized chromatin accessibility peak counts for 55,710 candidate cis-regulatory elements (CREs) across all 11 cell types. Cells are clustered based on cell type, sex, and race. (H) Gene ontology profiles of differentially active genes based on CREs in (G). (I) Row-normalized motif enrichment (ChromVAR) z-scores for the 500 most variable transcription factor motifs, across cell type, sex, and race. Select motifs and corresponding transcription factors are highlighted. (J) Enrichment z-scores projected onto UMAP coordinates of accessibility for select motifs from (I) (left panel). Normalized RNA expression projected onto UMAP profiles of scRNAseq profiles of islet cells as shown in Fig. 1A (right panel). Non-diabetic females: 9, non-diabetic males: 6. Differentially accessible chromatin peak counts have FDR adjusted q-value < 0.1, GO pathways have FDR adjusted q-value < 0.2. A Hypergeometric statistical test (clusterProfiler) used for over representation analysis in 3 h. All replicates are biological in this figure ($n = 15$, non-diabetic males: 6 females: 9).

incretin and KCl compared to islets from non-diabetic donors (Appendix Fig. 5A,B), without evidence for sex difference. T2D islets exhibited no difference in α cell function in hypoglycemic conditions compared to non-diabetic donors (Appendix Fig. 5C,D).

We compared the transcriptional profile of male and female HPAP donors with T2D. In contrast with non-diabetic donors, where most sex-associated genes were related to sex chromosomes (Fig. 2C,D), islets from T2D donors exhibited multiple sex-specific differences in DEGs from sex chromosomes and autosomes (Fig. 6A). When comparing DEGs in β and α cells from male and female T2D donors, the largest and most significant changes were restricted to sex-linked genes (Fig. 6B). We next compared the transcriptional profile of male and female HPAP donors with T2D to that of non-diabetic TUID and HPAP donors (Appendix Fig. 1A). Notably, in comparison of T2D vs. non-diabetic β cells, females exhibited more DEGs from autosomes (721 upregulated and 1164 downregulated) than males (111 upregulated and 99 downregulated), with only 5.2% of DEGs shared across sex (Appendix Fig. 6C,D). Similarly, in comparison of T2D vs. non-diabetic α cells, females exhibited more DEGs from autosomes (589 upregulated and 1552 downregulated) than males (14 upregulated and 6 downregulated), with only 0.28% overlap (Fig. 6C,F). When comparing T2D vs. non-diabetic donors in other cell types, females also exhibited more autosomal DEGs than males (Fig. 6C). We determined enrichment of gene ontology terms in these genes, and female β and α cells exhibited reduced mitochondrial function and respiration pathways in T2D (Fig. 6E,G) while male β cells exhibited reduced hormone and insulin secretion pathways in T2D (Fig. 6E). Enrichment of ontology terms for other islet cells in females and males are shown in Appendix Fig. 6.

## Sex- and cell-specific differences in T2D-associated genetic risk

While sex-stratified genome-wide association studies (GWAS) have been performed for T2D, the specific cell-types contribution to disease risk at each disease-associated locus remain unknown. To address this, we performed genomic enrichment analyses of our snATAC-seq open chromatin regions in T2D, fasting glucose, and fasting insulin GWASs using LD score regression. All islet endocrine cell types showed significant genomic enrichment (FDR < 0.05) in both male and female T2D GWAS, suggesting

common endocrine-driven mechanisms at disease risk loci (Fig. 7A; Appendix Fig. 7A). Notably, macrophages, lymphocytes, and quiescent stellate cells only showed enrichment in the T2D male GWAS, suggesting a sex-based heterogeneity in the immune regulation of T2D risk. Also of interest, while all islet endocrine cell types showed significant genomic enrichment in both male and female T2D GWAS, islet endocrine cells showed significant genomic enrichment for fasting glucose GWAS in female only (Appendix Fig. 7A,B).

We also assessed whether sex-specific differentially accessible chromatin regions lie within T2D risk loci. In total, 40 regions that were differentially accessible across sex (FDR < 0.1) overlapped with variants from 37 unique T2D risk signals (Fig. 7B). One differentially accessible chromatin region, in particular, was only detected in female lymphocytes, with no detectable reads in male lymphocytes (b38; 19:19627169–1962913019) (Fig. 7C). The differentially accessible female lymphocyte region overlaps with 4 T2D variants at the TM6SF2 risk locus (index variant rs188247550). We found differentially accessible regions in male δ cells to overlap with T2D-associated variants in *GCK*, *KCNQ1*, *PIK3R1*, in contrast to females (Fig. 7C; Appendix Fig. 7C). We also found *GLI2* to overlap in female ductal cells, in contrast to males (Appendix Fig. 7D). Similarly, in the case of male endothelial cells, we found differentially accessible regions to overlap with variants regulating *HNF1A*, *NEUROG3*, and in case of acinar cells *SLC30A8* (Fig. 7C). Previously, 31 variants across 28 T2D risk loci were reported to have sex-specific effects on T2D in a trans-ancestry GWAS, including one variant near *TM6SF2* (rs8107974), two variants at *GLI2* (rs11688931, rs11688682), and one variant at *KCNQ1* (rs2237895) (Mahajan et al, 2018). Inclusion of two additional T2D meta-analyses which included the X-chromosome found no additional overlap in T2D risk loci with differentially accessible chromatin regions on the X-chromosome (Bonàs-Guarch et al, 2018; Vujkovic et al, 2020).

## Discussion

Our study provides a single-cell atlas of sex-specific genomic differences in pancreatic islet cell types in subjects with and without T2D. In non-diabetic islet cells, sex differences in sex-linked genes predominate. In females, *XIST* and its negative regulator *TSIX* are upregulated across all islet cells, suggesting a role of X-chromosome dosage compensation (Heard et al, 1997) in

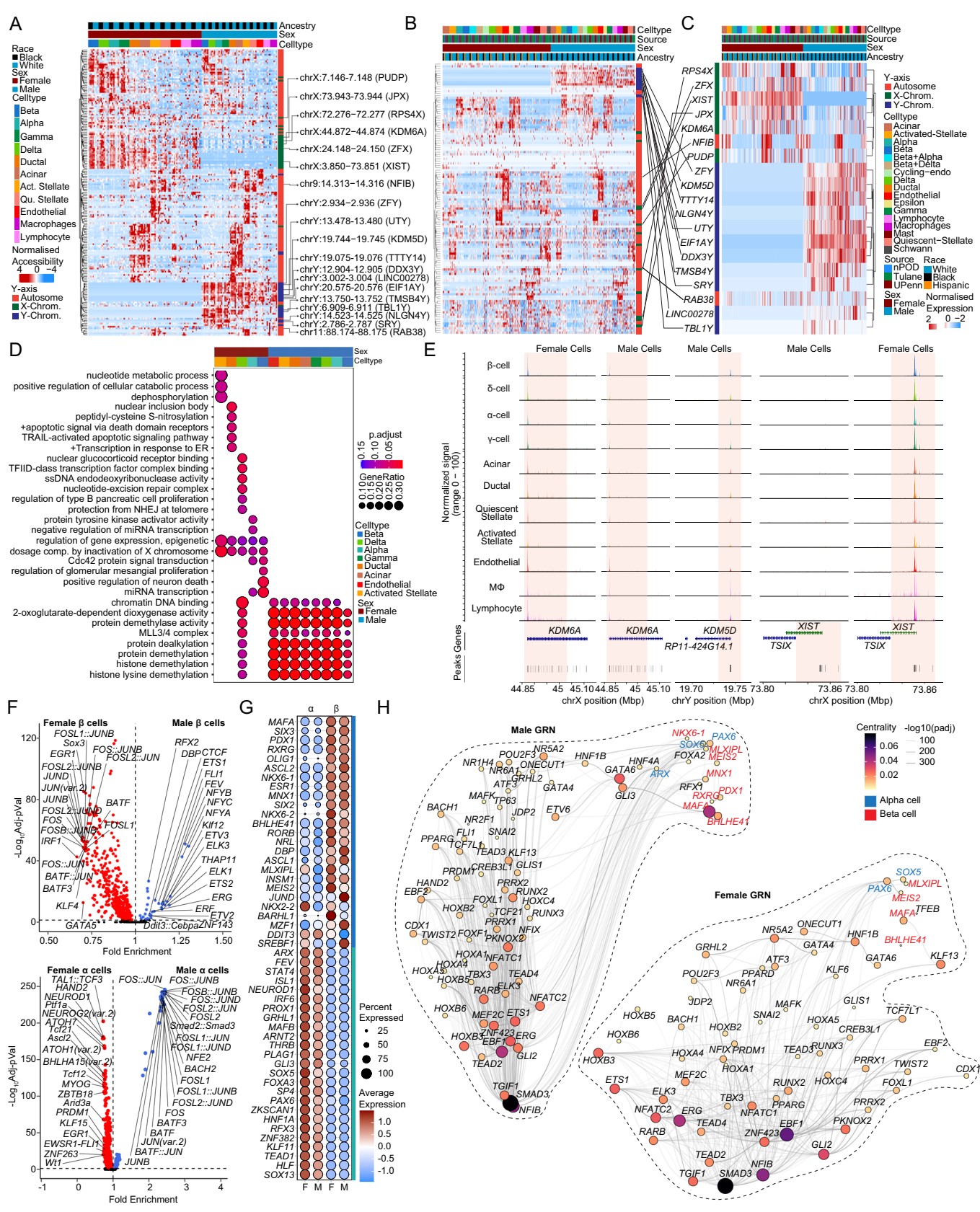

**Figure 4. Sex-based enrichment for sex-chromosome gene accessibility in human islet cells.**

(A) Row-normalized differentially accessible chromatin peaks across sex and cell-type. *XIST*, *KDM5D*, and *KDM6A* are highlighted. (B) Row normalized expression profiles for genes in a 100 kb boundary in proximity to cCREs corresponding to (A) in scRNAseq dataset. (C) Row normalized expression profiles for the subset of genes corresponding to (B) and differentially expressed genes across sex in scRNAseq dataset. (D) Gene ontology dotplot showing differential pathways active across multiple cell types based on sex. (E) Aggregated read density profile within a 50-kb window flanking a TSS for *KDM6A*, *KDM5D*, and *XIST*. (F) Violin plots of differentially accessible motifs identified using ChromVAR in female and male β cells (top) α cells bottom). (G) Dotplot across sex showing top 25 ranked differentially expressed transcription factors across beta and alpha cells. (H) Gene regulatory network UMAP embedding of pan-islet transcription factor (TF) activity, based on co-expression, and inferred interaction strength across TFs, for males (left) and females (right). Size/color represent PageRank centrality of each TF. TFs from (G) are highlighted for β (red) and α (blue) cell types. Differentially accessible chromatin peak counts have FDR adjusted q-value < 0.1, GO pathways have FDR adjusted q-value < 0.2. A Hypergeometric statistical test (clusterProfiler) used for over representation analysis in (D). A Wald statistical test (DEseq2) was used for (C–F). A Wilcoxon rank-sum test was used for (F). A generalized linear model with a Wald statistical test was utilized for (H). All tests utilized a Benjamini–Hochberg post hoc correction (FDR). For (A, D–H), $n = 15$ non-diabetic males: 6 females: 9. For (B, C), $n = 36$ non-diabetic females: 20, males: 16. All replicates in this figure are biological.

human islet function. Similarly, the Y-linked ubiquitin specific peptidase *USP9Y* (Lee et al, 2003) and S4 ribosomal protein *RPS4Y1* (Zinn et al, 1994) genes are expressed exclusively in all male cells, also suggesting a role for these genes in male islet function. Most genes on one X chromosome of XX cells are silenced in development through X chromosome inactivation by *XIST*, thus normalizing X chromosome genes dosage between sexes. However, some X chromosome genes escape inactivation and are expressed from both alleles in XX cells (Berletch et al, 2010; Tukiainen et al, 2017). These "X-escape genes" are conserved between mouse and humans, and several are epigenetic remodelers that promote histone modification to regulate genome access to transcription factors. For example, the histone demethylase *KDM6A* escapes X inactivation (Chen et al, 2012) and was more accessible and expressed in female β and α cells. *KDM6A* promotes sex differences in T cell biology (Itoh et al, 2019). Similarly, *KDM5D* is only expressed from the male Y chromosome and was overexpressed in male β and α cells. *KDM5D* drives sex differences in male osteogenesis, cardiomyocyte, and cancer (Gu and Chu, 2021; Li et al, 2016; Merten et al, 2022; Meyfour et al, 2019). Thus, sex differences in expression of chromatin remodelers like KDM6A or KDM5D may influence sex-specific chromatin access to transcription factors promoting sex differences in islet function. Consistent with this possibility, we observed a five-to-ten-fold greater number of transcription factor-specific accessible motifs in female compared to male α and β cells.

Non-diabetic female islets exhibited greater ATP-mediated respiration and coupling efficiency than those of males, which is consistent with females' mitochondria having greater functional capacity (Cardinale et al, 2018; Ventura-Clapier et al, 2017). In contrast, female β cells from T2D donors showed reduced activation of pathways enriched in mitochondrial function compared to female β cells from non-diabetic donors, which was not observed in male β cells. In addition, in comparison of T2D vs. non-diabetic β cells, females exhibited seven to ten-times more dysregulated autosomal genes than males. Taken together this suggests that females β cells are resilient and must develop more severe dysfunction to fail than those of males. This is consistent with the observation that female mouse islets retain greater β cell function during metabolic stress (Brownrigg et al, 2023). Thus, in the transition from normoglycaemia to T2D, female β cell develop greater mitochondrial dysfunction than those of males. This may explain why males are more prone to β cell failure than females as discussed in the introduction. Sex hormones may explain these differences, as estrogen and

androgen receptors affect mitochondrial function in female and male β cells (Xu et al, 2023a; Zhou et al, 2018). However, since differences between islets from non-diabetic and T2D donors were present outside of the in vivo hormonal environment, cell autonomous factors, such as the sexually dimorphic sex chromosomes genes described above are more likely to be involved in these differences.

We find little evidence of differences across race/ethnicity, although inflammatory cytokine signaling was increased in black male β cells via *IL18*, a cytokine implicated in diabetes, obesity, and metabolic syndrome (Harms et al, 2015; Trøseid et al, 2010; Zaharieva et al, 2018). In addition, non-diabetic black male islets exhibit decreased cAMP-stimulated insulin secretion compared to white male islets. This is reminiscent of ketosis-prone diabetes, a form of T2D mostly observed in males of sub-Saharan African descent with severe β cell failure (Louet et al, 2008; Mauvais-Jarvis et al, 2004; Umpierrez et al, 2006).

In genomic enrichment analyses of our snATAC-seq open chromatin regions for T2D GWAS, we find that differentially accessible regions overlap with T2D-associated variants in a sex- and cell-specific manner. One accessible chromatin region in female lymphocytes overlaps with 4 T2D-associated variants at the TM6SF2 risk locus and was not detectable in male lymphocytes. Previously, 31 variants across 28 T2D risk loci were reported to have sex-specific effects on T2D in a trans-ancestry GWAS, including one variant near the same TM6SF2 locus (Mahajan et al, 2018). We also found differentially accessible regions to overlap with classical T2D variants in male but not female δ cells (*GCK, KCNQ1,* and *PIK3R1*), endothelial cells (*HNF1A* and *NEUROG3*), and acinar cells (*SLC30A8*). Surprisingly no region overlapped with T2D variants in β cells. Surprisingly, all islet endocrine cells showed significant genomic enrichment in female only for fasting glucose GWAS. This again suggests sex differences in the pathogenesis of hyperglycaemia.

A strength of our study is the use of 'pseudo-bulk' profiles aggregated per cell type in each sample. Collapsing cell profiles by sample enables to effectively control for pseudo-replication due to cells being sampled from a fixed number of donors, whereas treating each cell from the same cluster as an independent observation leads to inflated *p*-value and spurious results. This approach has demonstrated high concordance with bulk RNA-seq, proteomics and functional gene ontology data (Heumos et al, 2023; Squair et al, 2021). We applied a hypergeometric statistical model using 'pseudo-bulk' count data correcting for library composition bias and batch effects in the scRNA-seq (Elgamal

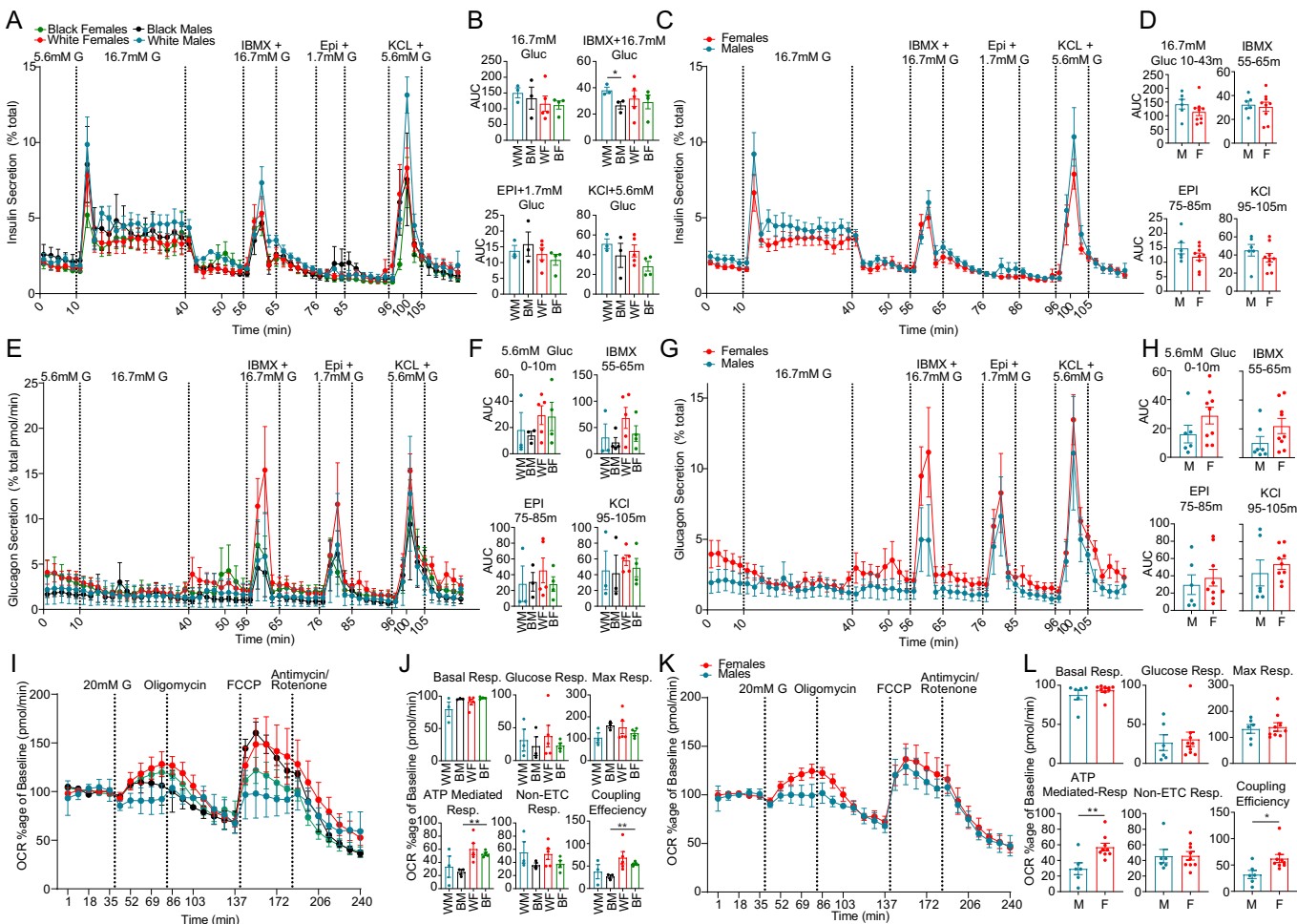

**Figure 5. Sex and race differences in islet hormone secretion and bioenergetics.**

(A) Dynamic insulin secretion assay, showing response to 16.7 mM glucose, IBMX + 16.7 mM Glucose, epinephrine + 1.7 mM Glucose, and potassium chloride + 5.6 mM glucose. Each curve represents secretion normalized to total insulin content across sex and race. (B) Area under the curve (AUC) measurements for incretin driven insulin secretion measurements outlined in (A). (C) Dynamic insulin secretion assay, showing response to 16.7 mM glucose, IBMX + 16.7 mM Glucose, epinephrine + 1.7 mM Glucose, and potassium chloride + 5.6 mM glucose. Each curve represents secretion normalized to total insulin content across sex. (D) Area under the curve (AUC) measurements for incretin driven insulin secretion measurements outlined in (B). (E) Dynamic glucagon secretion assay, showing response to 16.7 mM glucose, IBMX + 16.7 mM Glucose, epinephrine + 1.7 mM Glucose and potassium chloride + 5.6 mM glucose. Each curve represents secretion normalized to total glucagon content across sex and race. (F) Area under the curve (AUC) measurements for incretin driven insulin secretion measurements outlined in (E). (G) Dynamic glucagon secretion assay, showing response to 16.7 mM glucose, IBMX + 16.7 mM Glucose, epinephrine + 1.7 mM Glucose, and potassium chloride + 5.6 mM glucose. Each curve represents secretion normalized to total glucagon content across sex. (H) Area under the curve (AUC) measurements for incretin driven insulin secretion measurements outlined in (G). (I) Oxygen consumption ratio for islets across sex and race. (J) Basal respiration, glucose mediated respiration, maximal (max) respiration, ATP-mediated respiration, non-electron transport chain (ETC) respiration and coupling efficiency, across sex and race. (K) Oxygen consumption ratio for islets across sex. (L) Basal respiration, glucose mediated respiration, maximal (max) respiration, ATP-mediated respiration, non-electron transport chain (ETC) respiration and coupling efficiency, of human islets across sex. Data is shown as mean ± standard error (SE), bars SE. *p*-values for (B) = 0.0354, (J) = 0.0011, (L) = 0.0065. An unpaired *t*-test was used to perform statistics for (B, D, F, H, J and L). All replicates are biological in this figure (*n* = 15, non-diabetic males: 6 females: 9). Source data are available online for this figure.

et al, 2023). This approach has enabled us to recapitulate biological ground truth, where we demonstrate high concordance between accessible chromatin and associated active genes across human islet cells.

In conclusion, this study establishes an integrated accessible chromatin and transcriptional map of human islet cell types across sex and race at single-cell resolution, reveals that sex-specific genomic differences in non-diabetic individuals predominantly through sex chromosome genes, and reveals genomic differences in

islet cell types in T2D which highlights mitochondrial failure in females.

## Limitations of the study

Despite the inclusion of seven black donors (Tulane dataset) to promote genetic diversity, our study is limited by the small sample size. Future extramural funding for the inclusion and study of diverse genetic datasets is essential. Another key consideration is library

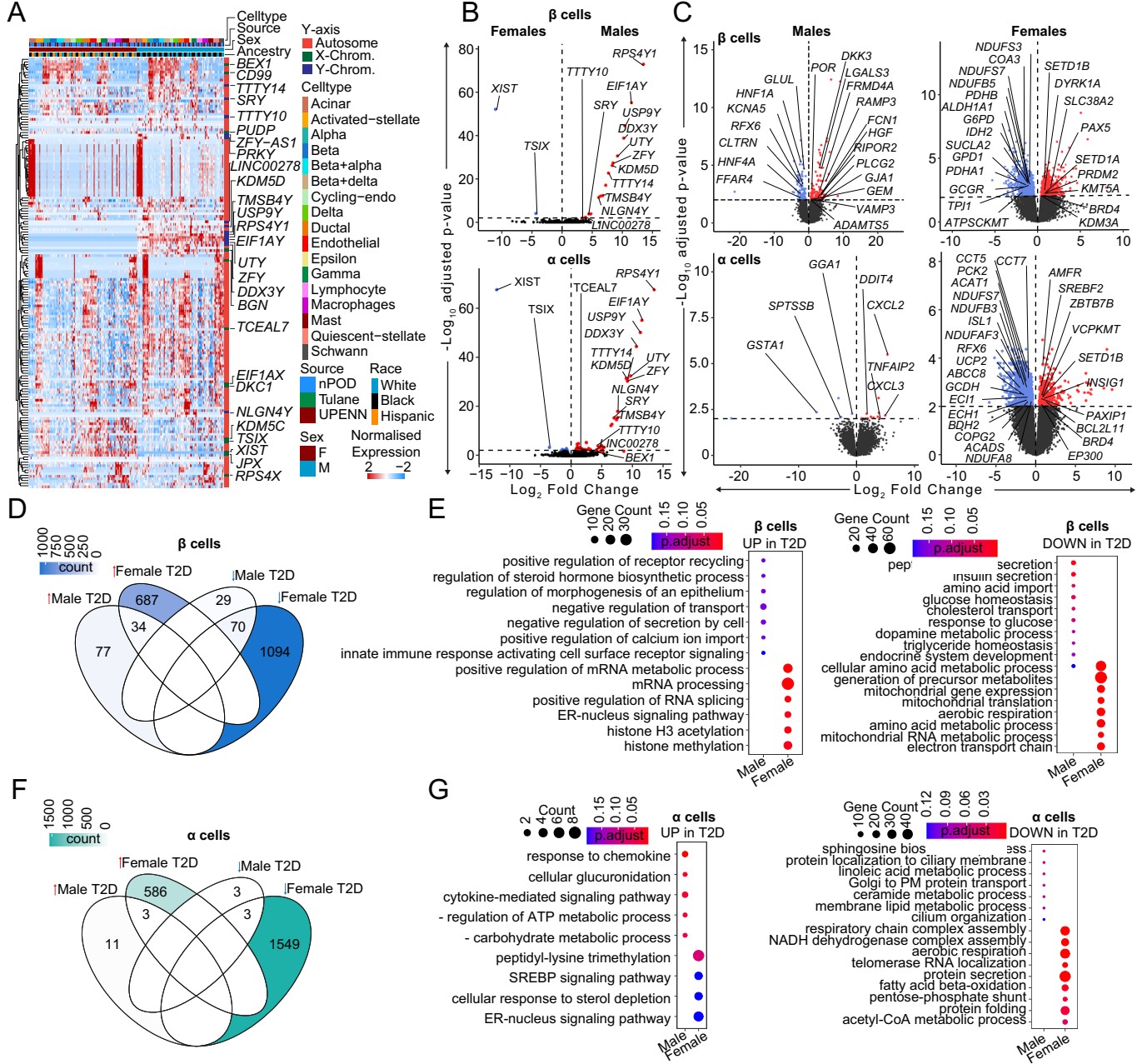

**Figure 6. Transcriptional differences in T2D compared to non-diabetic endocrine cells.**

(A) Heatmap of DEGs across T2D donors. (B) Violin plots showing DEGs across male and female T2D β/α cells. (C) Violin plots showing DEGs across β/α cells when diabetic donors are compared to non-diabetic controls across sex. (D) Venn diagram showing DEGs across different sex-disease comparisons in case of β cells. Color denotes the number of genes. (E) Gene ontology dotplot for upregulated and downregulated pathways for β-cell DEGs. (F) Venn diagram showing DEGs across different sex-disease comparisons in case of α cells. Color denotes the number of genes. (G) Gene ontology dotplot for upregulated and downregulated pathways for α-cell DEGs. DEGs have FDR adjusted q-value < 0.01, GO pathways have FDR adjusted q-value < 0.2. A Wald statistical test (DEseq2) was used for (B, C). A Hypergeometric statistical test (clusterProfiler) used for over representation analysis in (E) and (G). All tests utilized a Benjamini–Hochberg post hoc correction (FDR). For (B, C) ($n = 52$, non-diabetic females: 20, males: 16, T2D females: 9, T2D males: 7). All replicates in this figure are biological.

composition bias owing to targeted islet sequencing, which is not a representation of all pancreatic cells, cell subtypes, or spatiotemporal domains (Domínguez-Bendala et al, 2021; Qadir et al, 2020). Even after utilizing a stringent ambient RNA correction methodology, invariably residual contaminant RNA can be observed across cells.

Emphasis is given on generating tools to adjust for ambient RNA particularly in case of pancreatic cells containing high expression of genes such as *INS* and *PRSS1*. The snATAC-seq is examining open enhancers within short proximity of genes, but distal enhancers are also essential for β cell function and could be implicated in T2D.

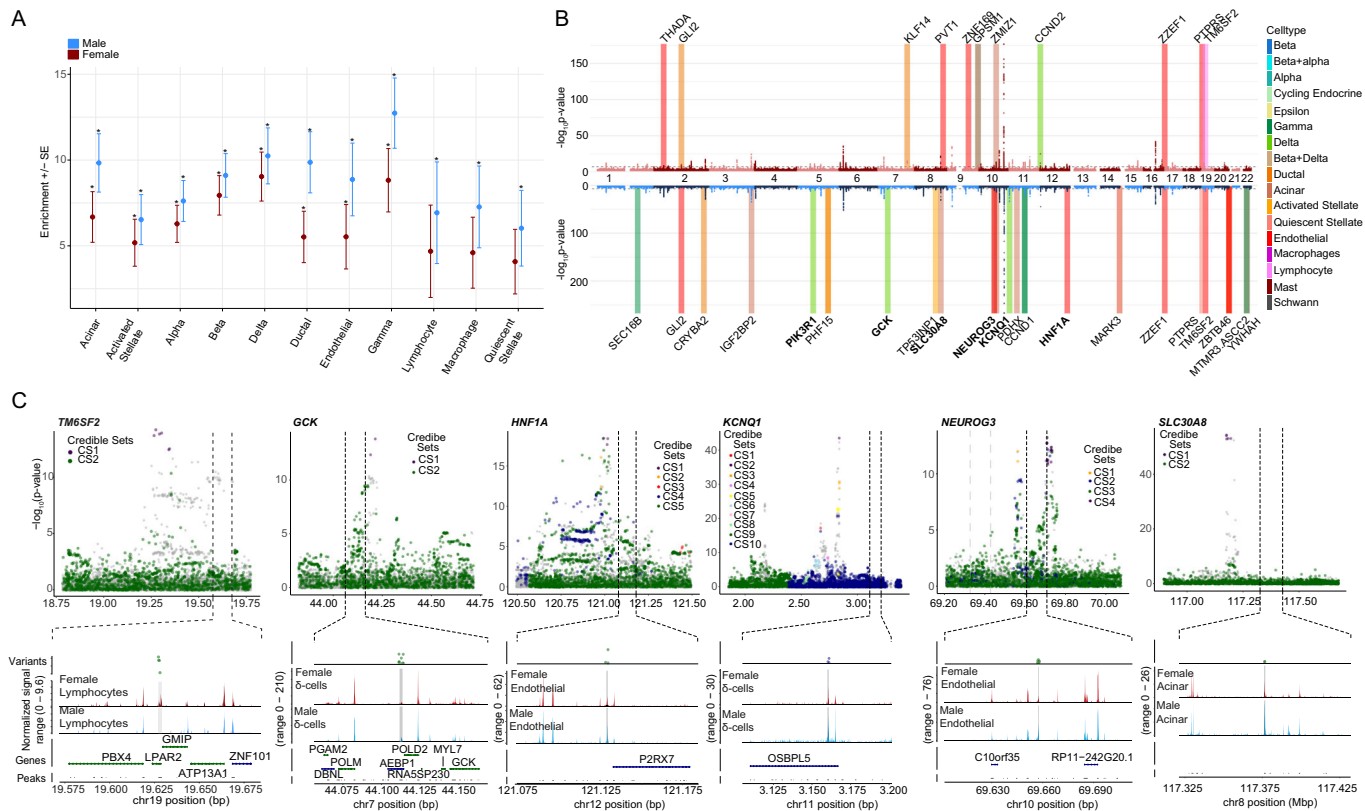

**Figure 7.  GWAS utilizing the DIAMANTE Type 2 diabetes dataset shows cell type and sex-specific variants influencing Type 2 diabetes risk.**

(**A**) Cell-type genomic enrichment in male (*N* = 75,676) and female (*N* = 52,842) T2D GWAS using LD score regression. Error bars represent enrichment standard error in each sex. Significant enrichment was determined for each sex independently (*FDR < 0.05). (**B**) Miami plot of female (top) and male (bottom) sex-stratified DIAMANTE T2D GWAS. Differentially accessible peaks from snATACseq analysis for each sex are represented on their respective Manhattan plots. Bolded loci have evidence of sex-heterogeneity in DIAMANTE T2D GWAS. (**C**) Differentially accessible chromatin peaks at b38; 19:19627168–19629130 in female lymphocytes overlaps with credible set variants at T2D risk locus with index variant rs188247550, while in males differentially accessible chromatin peaks overlap with T2D risk loci at b38; 7:44111586–44113624 (*GCK*, rs116913033), b38; 11:3159900–3161041 (*KCNQ1*, rs445084) in delta cells, b38; 8:117376094–117376998 (*SLC30A8*, rs80244329) in acinar cells, b38; 10:69657288–69657771 (*NEUROG3*, rs61850200 and rs41277236) and b38; 12:121128766–121129441 (*HNF1A*, rs28638142) in endothelial cells. Gray bars showing index variants overlapping with differentially accessible regions. Data is shown as mean ± standard error (SE), bars SE. FDRs (adjusted *p*-value) for (**A**), (female) acinar = 1.65E−4, activated stellate = 2.04E−3, alpha = 1.44E−6, beta = 3.89E−9, delta = 3.29E−8, ductal = 2.63E−3, endothelial = 1.58E−2, gamma = 2.98E−5, and (male) acinar = 2.64E−7, activated stellate = 1.08E−4, alpha = 1.7E−8, beta = 1.81E−10, delta = 9.9E−9, ductal = 4.86E−7, endothelial = 1.95E−4, gamma = 1.43E−8, lymphocyte = 4.68E−2, macrophage = 7.7E−3, quiescent stellate = 2.16E−2. A Linkage Disequilibrium Score Regression (LDSC) statistical framework was used which utilizes regression models on comparing LD scores of credible sets and chi-squared statistics from GWAS results in (**A–C**). All replicates in this figure are biological.

# Methods

### Reagents and tools table

| Reagent/Resource | Reference or Source | Identifier or Catalog Number |
|---|---|---|
| **Chemicals, Enzymes and other reagents** | | |
| Phenol-red free RPMI medium | Gibco | 11835030 |
| Fetal Bovine Serum, charcoal stripped | Gibco | 12676029 |
| HEPES | Gibco | 15630080 |
| Sodium Pyruvate | Gibco | 11360070 |
| 2-Mercaptoethanol | Gibco | 31350010 |
| GlutaMAX | Gibco | 35050061 |
| Penicillin-Streptomycin | Gibco | 15140122 |
| Bio-Gel P-4 | Bio-Rad | 1504120 |

| Reagent/Resource | Reference or Source | Identifier or Catalog Number |
|---|---|---|
| IBMX | Sigma-Aldrich | I7018 |
| Epinephrine | Sigma-Aldrich | E4642 |
| **Software** | | |
| CellRanger | 10x Genomics | v4.0.0 |
| CellRanger ATAC | 10x Genomics | v1.2.0 |
| R | R Consortium | v4.3.2 |
| Python | Python Org | v3.13.0 |
| Seurat | Satija Lab | v4.3.0 |
| Signac | Satija and Stuart Labs | v1.10.0 |
| SoupX | Young Lab | v1.6.1 |
| DoubletFinder | Gartner Lab | v2 |

| Reagent/Resource | Reference or Source | Identifier or Catalog Number |
|---|---|---|
| DESeq2 | Love Lab | v1.36.0 |
| chromVAR | Greenleaf Lab | v1.22.1 |
| Harmony | Raychaudhuri Lab | v0.1.1 |
| MACS2 | Liu Lab | v3.0.2 |
| JASPAR 2020 | MRC | v0.99.8 |
| clusterProfiler | He Lab | v4.4.4 |
| Other | | |
| Mercodia Insulin ELISA kit | Mercodia | 10-1113-01 |
| Mercodia Glucagon ELISA kit | Mercodia | 10-1271-01 |
| 10x Chromium Controller | 10x Genomics | |
| Illumina NextSeq 2000 | Illumina | |
| Agilent 2100 Bioanalyzer | Agilent | |
| Qubit 2.0 fluorometer | ThermoFisher | |

## Methods and protocols

### Human pancreatic islets

De-identified human pancreatic islets from 15 male and female donors were obtained from PRODO Laboratories Inc, and the Integrated Islet Distribution Program (IIDP). The islets were left in culture at 37 °C in a humidified incubator containing 5% $CO_2$ overnight before any experiments were performed. Islets were cultured in phenol-red free RPMI medium (Gibco) containing 11 mM glucose, supplemented with 10% Charcoal Stripped FBS (Invitrogen), HEPES (10 mM; Gibco), Sodium Pyruvate (1 mM; Gibco), β-mercaptoethanol (50 μM; Invitrogen), GlutaMAX (2 mM; Gibco), and Penicillin-Streptomycin (1x; Gibco).

### Studies involving human cadaveric tissue

Samples originate from de-identified cadaveric donors and are institutional review board exempt.

### Measurement of insulin secretion in perifusion

Perifusion experiments were performed in Krebs buffer containing 125 mM NaCl, 5.9 mM KCl, 1.28 mM $CaCl_2$, 1.2 mM $MgCl_2$, 25 mM HEPES, and 0.1% bovine serum albumin at 37 °C using a PERI4-02 machine (Biorep Technologies). Fifty hand-picked human islets were loaded in Perspex microcolumns between two layers of acrylamide-based microbead slurry (Bio-Gel P-4, Bio-Rad Laboratories). For experiment 1, cells were challenged with either low or high glucose (5.6 mM or 16.7 mM), IBMX (100 μM), epinephrine (1 μM) or potassium chloride (20 mM) at a rate of 100 μL/min. After 60 min of stabilization in 5.6 mM glucose, cells were stimulated with the following sequence: 10 min at 5.6 mM glucose, 30 min at 16.7 mM glucose, 15 min at 5.6 mM glucose, 5 min at 100 μM IBMX + 16.7 mM glucose, 15 min at 5.6 mM glucose, 5 min at 1 μM epinephrine + 1.7 mM glucose, 15 min at 5.6 mM glucose, 15 min at 20 mM KCl + 5.6 mM glucose, and 15 min at 5.6 mM glucose. In case of experiment 2, islets were challenged with either low or graded high concentrations of glucose (2, 5, 11, or 20 mM) or potassium chloride (20 mM) at a rate of 100 μL/min. After 60 min of stabilization in 2 mM glucose, islets

were stimulated in the following sequence: 10 min at 2 mM glucose, 10 min at 7 mM glucose, 10 min at 11 mM glucose, 10 min at 20 mM glucose, 15 min at 2 mM glucose, 10 min at 20 mM KCl + 2 mM glucose, 10 min at 20 mM KCl + 11 mM glucose and, 10 min at 2 mM glucose. Samples were collected every minute on a plate kept at <4 °C, while the perifusion solutions and islets were maintained at 37 °C in a built-in temperature controlled chamber. Insulin and glucagon concentrations were determined using commercially available ELISA kits (Mercodia). Total insulin and glucagon release was normalized per total insulin or glucagon content respectively using a human insulin or glucagon ELISA kit (Mercodia).

For samples used as a part of the HPAP dataset, sample metadata and perifusion data were downloaded from the HPAP website: https://hpap.pmacs.upenn.edu/, for samples used as a part of this study. Data were organized based on insulin and glucagon secretion where available and plotted across sex.

### Bioenergetics

Islets were washed once with assay buffer (made from Agilent Seahorse XF Base Medium supplemented with 3 mM glucose and 1% charcoal striped FBS). Around 150 islets were transferred to each well of Seahorse XF24 Islet Capture Microplate (Agilent) and were incubated in assay buffer at 37 °C for 60 min before being transferred to Agilent Seahorse XFe24 Analyzer. Islets were maintained in the assay medium throughout the experiment, while oxygen consumption rate (OCR) and extracellular acidification rate (ECAR) were measured at basal (3 mM), glucose-stimulated level (20 mM) and after addition of oligomycin, carbonyl cyanide-4 (trifluoromethoxy) phenylhydrazone (FCCP), rotenone/antimycin according to manufacturer's instructions.

### Single-cell RNA indexing and sequencing

Human islets (500 IEQ per condition) were cultured overnight in a humidified incubator containing 5% $CO_2$ at 37 °C. Islet cells were then dispersed using TrypLE (Thermofischer), and immediately evaluated for viability (90.61 ± 3.04%) by Cellometer Automated Cell Counter (Nexcelom Bioscience) prior to single cell RNAseq library preparation. For 10x single cell RNAseq library preparation, 5000–6500 individual live cells per sample were targeted by using 10x Single Cell 3' RNAseq technology provided by 10x Genomics (10X Genomics Inc). Briefly, viable single-cell suspensions were partitioned into nanoliter-scale Gel Beads-In-EMulsion (GEMs). Full-length barcoded cDNAs were then generated and amplified by PCR to obtain sufficient mass for library construction. Following enzymatic fragmentation, end-repair, A-tailing, and adapter ligation, single cell 3' libraries comprising standard Illumina P5 and P7 paired-end constructs were generated. Library quality controls were performed by using Agilent High Sensitive DNA kit with Agilent 2100 Bioanalyzer (Agilent) and quantified by Qubit 2.0 fluorometer (ThermoFisher). Pooled libraries at a final concentration of 750 pM were sequenced with paired-end single index configuration by Illumina NextSeq 2000 (Illumina).

### Single-cell gene expression mapping

For the Tulane dataset, we utilized CellRanger v4.0.0 software using the [-mkfastq] command to de-multiplex FASTQ data. Reads were mapped and aligned to the human genome (10X genomics pre-built GRCh38-2020-A Homo sapiens reference transcriptome assembly) with STAR (95.33 ± 0.75% of reads confidently mapped to the human

genome) (Dobin et al, 2013). Subsequently, final digital gene expression matrices and c-loupe files were generated for downstream multimodal analysis. In case of the HPAP dataset we isolated data processed as described previously (nPod data: 87.91 ± 11.56 and UPenn 90.62 ± 5.44% of reads map confidently to genome) (Elgamal et al, 2023). Cellranger identified 75,619 (Tulane), 73,472 (nPOD) and 52,357 (UPenn) correctly allocated barcodes (cells), having 78,584 ± 40,590 (Tulane), 130,993 ± 289,368 (nPOD), 63,949 ± 29,598 (UPenn) reads/cell and 26,866 ± 680 (Tulane), 24,739 ± 8983 (nPOD), 24,183 ± 1254 (UPenn) genes/cell.

### Preliminary filtering and S4 R object creation

We deployed Seurat v4.3.0 (Hao et al, 2021; Stuart et al, 2019) scripts to perform merging, thresholding, normalization, principal component analysis (linear dimensionality reduction), clustering analysis (non-linear multidimensional reduction), visualization and differential gene expression analysis. Cells having total mitochondrial RNA contribution beyond 20% were eliminated from the analysis, along with cells expressing less than 500 or greater than 8000 total genes.

### Ambient RNA correction and doublet annotation

In droplet-based scRNAseq technologies, extracellular RNA from cells with compromised membrane integrity contaminates single-cell libraries (Heumos et al, 2023). This remains a challenge for pancreatic cells, as endocrine and exocrine cells are rich in select secreted RNA species. We used SoupX 1.6.1 (Young and Behjati, 2020) on raw feature barcode matrices correcting for ambient RNA across all 52 donors. Raw counts were corrected using SoupX and rounded to the nearest integer. As the TUID is not doublet corrected, we utilized DoubletFinder v2 (McGinnis et al, 2019) expecting 5% doublets, eliminating them from the dataset.

### Data normalization and clustering

SoupX corrected matrices were metadata annotated, and geometrically normalized (log10) at a scale factor of 10,000. The variance stabilization method (vst) method was used to find 2000 most variable features, which were later used for scaling and principal component analysis (PCA) using 20 components. and dimensions (UMAP). We batch corrected the datasets using Harmony 0.1.1 (Korsunsky et al, 2019), using donor library identity, 10X genomics chemistry (v2 or v3) and tissue source (Tulane, nPOD or UPenn) as covariates in the batch model. Uniform manifold approximation and projection (UMAP) and neighbors were calculated using Seurat v4.3.0 (Hao et al, 2021; Stuart et al, 2019). Finally, we hyperclustered data using a Leiden algorithm at a resolution of 6. We observed poor quality cells to remain in the dataset (low relative total RNA and gene counts yet within threshold), and excluded these from the analysis, and performed re-clustering as described above. Finally, we assigned identities to clusters based on pancreatic cell-specific gene sets (Qadir et al, 2020; Van Gurp et al, 2022), resulting in 17 discrete clusters, totaling 141,739 high-quality cells.

### Cell type-specific marker genes

Statistical approaches to define DEGs across cell types using aggregated "pseudobulked" RNA count data, out-perform single-cell DEG models (Heumos et al, 2023; Murphy and Skene, 2022; Squair et al, 2021). Infact, pseduobulk DEG methods demonstrate the highest Mathews Correlation Coefficient, a balanced machine learning performance testing model, capable of evaluating models classifying binary data (Chicco and Jurman, 2020; Murphy and

Skene, 2022). Therefore, we performed an unbiased differential analysis of cell cluster-specific marker genes using the [FindAll-Markers] function in Seurat. We employed DESeq2 v1.36.0 (Love et al, 2014) to perform DEG testing, where a cluster must express a gene in at least 25% of cells, have a 2x fold difference, and a Benjamini–Hochberg FDR adjusted *p*-value < 0.01 (α = 1%). Aggregated counts were compared across cell types and donors.

### Sex, race, and disease type specific marker genes

Based on facts outlined above, we employ a previously described statistical model (Elgamal et al, 2023) using DESeq2 v1.36.0 (Love et al, 2014) to evaluate statistical differences across human islet cell types based on race, sex, and disease, metadata profiles across donors. A DEG is defined as a gene having a Benjamini–Hochberg adjusted *p*-value < 0.1 (α = 10%).

### Single nuclear assay for transposase-accessible chromatin indexing and sequencing

Human islets (500 IEQ per condition) were cultured overnight in a humidified incubator containing 5% $CO_2$ at 37 °C. Islet cells were then dispersed using TrypLE (Thermofischer), and immediately evaluated for viability (90.61 ± 3.04%) by Cellometer Automated Cell Counter (Nexcelom Bioscience) prior to single nuclei ATAC library preparation. Nuclei were isolated based on the 10X genomics Nuclei isolation protocol (CG00169 Rev D) with some modifications. We observe that the usage of 0.5 ml tubes yields superior nuclei collection. Furthermore, we optimize based on a sample-to-sample basis the time for cell lysis (3–5 min). The final lysis buffer concentration for Nonidet P40 was 0.15% over the 0.1% recommendation. Finally, in addition to the final wash with wash buffer, we perform a final wash with the 10X Genomics Nuclei Buffer (PN-2000153/2000207). Nuclei are always kept <0 °C, visually inspected for integrity and quality using a viability dye, prior to library prep which was performed within 30 min. Briefly, 5000–6500 isolated nuclei were incubated with a transposition mix to preferentially fragment and tag the DNA in open regions of the chromatin. The transposed nuclei were then partitioned into nanoliter-scale Gel Bead-In-emulsions (GEMs) with barcoded gel beads, a master mix, and partition oil on a chromium chip H. Upon GEM formation and PCR, 10x barcoded DNA fragments were generated with an Illumina P5 sequence, a 16nt 10x barcode, and a read 1 sequence. Following library construction, sequencing-ready libraries were generated with addition of P7, a sample index, and a read 2 sequence. Quality controls of these resulting single-cell ATAC libraries were performed by using Agilent High Sensitive DNA kit with Agilent 2100 Bioanalyzer (Agilent) and quantified by Qubit 2.0 fluorometer (ThermoFisher). Pooled libraries at a final concentration of 750pM were sequenced with paired-end dual indexing configuration by Illumina NextSeq 2000 (Illumina) to achieve 40,000–30,000 read pairs per nucleus.

### Single nuclei accessible chromatin mapping

We utilized CellRanger ATAC v1.2.0 software using the [-mkfastq] command to de-multiplex FASTQ data. Reads were mapped and aligned to the human genome (10X genomics pre-built GRCh38-2020-A Homo sapiens reference transcriptome assembly) with STAR (70.70 ± 11.46% of reads confidently mapped to the human genome) (Dobin et al, 2013). Cellranger identified 84,741 correctly annotated barcodes (cells), having an average transcriptional start

site (TSS) enrichment score of $6.27 \pm 1.38$ and $73.55 \pm 6.78\%$ fragments overlapping peaks/sample. We then utilized Signac's peak calling tool to call peaks on our dataset using MACS2 (Zhang et al, 2008). We utilize the [CallPeaks()] function to annotate accessible peaks using MACS2.

### Preliminary filtering and S4 R object creation

We deployed Seurat v4.3.0 (Hao et al, 2021; Stuart et al, 2019) coupled with Signac v1.10.0 (Stuart et al, 2021) scripts to perform merging, thresholding, normalization, principal component analysis (linear dimensionality reduction), clustering analysis (non-linear multidimensional reduction), visualization and differential gene expression analysis. Cells having a TSS enrichment score of <2, peak region fragments less than 2000 or more than 20,000 counts, percentage reads in peaks <30%, blacklist ratio >0.05, nucleosome ratio >4 and, fraction reads in promoters <0.2 were eliminated from the analysis.

### Doublet annotation

It is increasingly challenging to detect multiplets in droplet-based snATAC data, owing to sparsity and low dynamic range. We employed AMULET (Thibodeau et al, 2021) within the scDblFinder v1.10.0 (Germain et al, 2022) R package on raw fragment barcode matrices correcting for all 15 donors, using the authors recommendations.

### Data normalization and clustering

We used a unified set of peaks across all 15 datasets, annotating genes using EnsDb.Hsapiens.v86 (Rainer, 2017). We estimated gene activity using Signac's GeneActivity function, by extracting gene coordinates and extending them to include the 2 kb upstream region, followed by geometric normalization (log10). We next performed non-linear multidimensional reduction using term frequency-inverse document frequency (TF-IDF) weighted peak counts transformed to binary data. Weighted data was reduced to 30 dimensions using RunSVD function. We batch corrected the datasets using Harmony 0.1.1 (Korsunsky et al, 2019) using 30 nearest neighbors, using donor library identity as a covariate in the batch model. The first singular value decomposition (SVD) component correlated with read depth and was eliminated from UMAP projection dimensionality reduction, and SLM (Waltman and Van Eck, 2013) clustering, based on recommendations provided in Signac.

Upon performing iterative clustering and after removing low-quality cells, we end up with 52,613 nuclei having 255,194 peak features spanning 11 clusters. We classified clusters based on described gene activities across islet cells (Chiou et al, 2021), followed by validating identity with label transfer, from our RNAseq atlas dataset using the FindTransferAnchors function. Finally, we stored an additional modeled predicted RNA expression matrix within the snATAC object using the TransferData function.

### Cell type specific marker genes

To evaluate differentially accessible regions (DARs) we used a Wilcoxon rank-sum test comparing a cluster of cells against all other clusters, defining DARs as those peaks expressed in at least 5% of cells, having a foldchange >2, Benjamini–Hochberg FDR adjusted $p$-value < 0.05 ($\alpha = 5\%$) and restricting to those peaks that are within a 100 kb window of a gene.

### Sex, race, and disease type specific marker genes

In order to evaluate population wide differences, we employed the similar model utilized for scRNAseq (Elgamal et al, 2023). A DAR is defined as a peak having a Benjamini–Hochberg adjusted $p$-value < 0.1 ($\alpha = 10\%$).

### Single-cell motif enrichment

We used chromVAR v1.22.1 (Schep et al, 2017) to estimate transcription factor motif enrichment z-scores across all cells. We used a peak by cell sparse binary matrix correcting for GC content bias based on the hg38 genome (BSgenome.Hsapiens.UCSC.hg38). We use the non-redundant JASPAR 2020 core vertebrate motif database (Fornes et al, 2019) calculating bias-corrected deviation z-scores across single cells. We then calculated average transcription factor motif enrichment z-scores across single cells in a cluster. We used aggregate cell average z-scores to evaluate differentially accessible motifs (DAMs) across clusters, using a Benjamini–Hochberg FDR corrected $p$-value < 0.05.

### Gene set enrichment and pathway analysis

In order to perform gene set enrichment analysis (GSEA) (Subramanian et al, 2005), we downloaded the entire molecular signatures database (MSigDB) v3 (Liberzon et al, 2011; Subramanian et al, 2005) for C5 human gene ontological terms, using clusterProfiler v4.4.4 (Wu et al, 2021) or using an R based deployment (https://github.com/wjawaid/enrichR) of EnrichR (Kuleshov et al, 2016). We subset the C5 database, restricting terms to biological processes and perform functional pathway annotation using the compareCluster function. We define a pathway to be statistically significant at a Benjamini–Hochberg FDR adjusted $p$-value < 0.2 ($\alpha = 20\%$). We performed functional pathway mapping using the cnetplot function.

### Gene regulatory network analysis

In order to infer gene regulatory networks (GRNs) we utilized Pando (Fleck et al, 2023) while using the predicted RNA expression profile and MACS2 components of our snATAC dataset while interrogating TFs for which motifs exist. The coefficients of Pando's model highlight a quantified measure of interaction across cCRE-TF pair and a downstream target gene, resulting in a regulatory graph which can be plotted using non-linear multidimensional reduction.

### Cell type-specific genomic enrichment (LDSC)

Sex-stratified T2D (DIAMANTE), fasting insulin (MAGIC), and fasting glucose (MAGIC) GWAS summary statistics were mapped using dbSNP 155 in order to add variant rsIDs (Lagou et al, 2021; Mahajan et al, 2018). Summary statistics were coerced into a standardized format using the Munge sumstats wrapper within LDSC (Bulik-Sullivan et al, 2015). Briefly, alleles were matched and subset to hapmap3 variants and a minor allele frequency threshold of greater than 0.01 was used. Functional annotations were generated for each of 11 cell type in the snATACseq object using cell type-specific peak annotations and 1000 Genomes Project European reference panel linkage disequilibrium. Linkage disequilibrium scores were calculated for functional annotations using a 1 centimorgan linkage disequilibrium window. Partitioned heritability was run between sex-stratified GWAS' and cell type annotations to calculate genomic enrichment (Finucane et al, 2015). Benjamini–Hochberg multiple test correction was used to correct

enrichment *p*-values for the total number of cell types tested and significance was determined by FDR < 0.05.

### Sex-specific chromatin accessibility on T2D risk

To assess whether sex-specific chromatin accessibility is shared with known T2D risk loci, we used bedtools intersect to determine whether sex-specific peaks across the 11 cell types in our snATACseq object harbored shared variants with previously computed T2D credible sets. Differentially accessible peaks across sex were determined using Seurat's FindMarkers function, as previously described (Hao et al, 2021). Peaks on the Y-chromosome were removed and multiple test correction was performed on the remaining peaks *p*-values using a Benjamini–Hochberg FDR. Peaks were considered differentially accessible in male samples if they had an average log2 fold change greater than 1 and an FDR < 0.1 and peaks were considered differentially accessible in female samples if they had an average log2 fold change less than 1 and an FDR < 0.1. For T2D risk loci, all variants within 99% credible sets were used in our analysis.

## Data availability

The datasets and computational pipelines produced in this study are available in the following databases: Single-cell RNA-sequencing data: Gene Expression Omnibus GSE266291. Single nuclei ATAC sequencing data: Gene Expression Omnibus GSE266405. A description of coding environments required to reproduce analysis in this paper are outlined in: https://github.com/FMJLabTulane/sex_regulome_pancreas. Data can be interactively viewed on using an online tool created using shinycell (Ouyang et al, 2021) for scRNAseq data (http://tools.cmdga.org:3838/islet-scrna-sex-differences/) for snATACseq data access view (http://tools.cmdga.org:3838/islet-scatac-sex-differences/) and a differential gene expression browser (tools.cmdga.org:3838/islet-deseq-sexes/), all hosted on the common metabolic disease genome atlas (CMDGA, raw rds files available here alongside GEO https://cmdga.org/publications/11e6314f-f303-485f-8b90-0ef22330545c/).

The source data of this paper are collected in the following database record: biostudies:S-SCDT-10_1038-S44318-024-00313-z.

## Peer review information

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

## Acknowledgements

This work was supported by National Institutes of Health grants DK074970 and P20GM152305 (FM-J), DK105554 (KJG), HG012059 (KJG), DK114650 (KJG), DK120429 (KJG), U.S. Department of Veterans Affairs Merit Award BX005812 (FM-J), and the Tulane Center of Excellence in Sex-Based Precision Medicine (FM-J). The preparation of human pancreatic islets provided by the Integrated Islet Distribution Program (IIDP) (RRID:SCR_014387) at City of Hope were funded by NIH grant 2UC4DK098085. This manuscript used data acquired from the Human Pancreas Analysis Program (HPAP-RRID:SCR_016202) Database (https://hpap.pmacs.upenn.edu/), a Human Islet Research Network (RRID:SCR_014393) consortium (UC4-DK-112217, U01-DK-123594, UC4-DK-112232, and U01-DK-123716).

## Author contributions

**Mirza Muhammad Fahd Qadir**: Conceptualization; Resources; Data curation; Software; Formal analysis; Validation; Investigation; Visualization; Methodology; Writing—original draft; Project administration; Writing—review and editing. **Ruth M Elgamal**: Resources; Data curation; Software; Formal analysis; Methodology; Writing—original draft. **Kejing Song**: Data curation; Writing—original draft. **Parul Kudtarkar**: Data curation; Formal analysis. **Siva S V P Sakamuri**: Data curation; Formal analysis. **Prasad V Katakam**: Supervision; Writing—review and editing. **Samir S El-Dahr**: Supervision; Writing—review and editing. **Jay K Kolls**: Data curation; Writing—review and editing. **Kyle J Gaulton**: Conceptualization; Resources; Software; Supervision; Funding acquisition; Investigation; Writing—review and editing. **Franck Mauvais-Jarvis**: Conceptualization; Supervision; Funding acquisition; Writing—original draft; Project administration; Writing—review and editing.

Source data underlying figure panels in this paper may have individual authorship assigned. Where available, figure panel/source data authorship is listed in the following database record: biostudies:S-SCDT-10_1038-S44318-024-00313-z.

## Disclosure and competing interests statement

The authors declare no competing interests.

