## [Peer Review File · The EMBO Journal]

Sex-specific regulatory architecture of pancreatic islets from subjects with and without type 2 diabetes

Franck Mauvais-Jarvis, Mirza Qadir, Ruth Elgamal, Kejing Song, Parul Kudtarkar, Siva Sakamuri, Prasad Katakam, Samir El-Dahr, Jay Kolls, and Kyle Gaulton

Corresponding author: Franck Mauvais-Jarvis (fmauvais@tulane.edu)

Review Timeline:

Submission Date:	5th Sep 24
Editorial Decision:	11th Oct 24
Revision Received:	15th Oct 24
Editorial Decision:	23rd Oct 24
Revision Received:	23rd Oct 24
Accepted:	4th Nov 24

Editor: Ieva Gailite

Transaction Report:

(Note: Please note that the manuscript was previously reviewed at another journal and the reports were taken into account in the decision making process at The EMBO Journal. With the exception of the correction of typographical or spelling errors that could be a source of ambiguity, letters and reports are not edited. Depending on transfer agreements, referee reports obtained elsewhere may or may not be included in this compilation. Referee reports are anonymous unless the Referee chooses to sign their reports.)

Point-by-point response to Reviewers

Reviewer #1 (Remarks to the Author):

This manuscript reports two potentially exciting observations concerning how sex could influence T2D mechanisms. If true they could change our views on T2D.

One observation is that males and females with T2D have different beta cell gene expression abnormalities. The evidence supporting this finding is not particularly strong, as discussed below.

RESPONSE: We thank the Reviewer for acknowledging our exciting observations and as discussed in more details below, the sex difference in T2D β cell autosomal gene expression is supported by stringent methodology and is a seminal observation that needs to be reported.

Another interesting observation is that sex-specific GWAS associations appear to act on active regions from different cell types, or on regions that show sex-specific accessibility differences.

RESPONSE: We thank the Reviewer and agree that this is an interesting observation

The description of sex-specific autosomal RNA/atac differences in beta/alpha cells in non-diabetic samples is also of interest, but could be analyzed in greater depth, providing more convincing evidence that they are truly sex-specific differences.

RESPONSE: The few sex-associated autosomal genes in α/β cells of non-diabetic donors exhibited a small effect size compared to sex-associated X and Y genes (Fig. 2c and d), which as suggested is discussed in the revised manuscript (Line 139-140).

Lastly, a more general comment, the first figures of the manuscript include descriptive information that is of limited interest to the reader. It is obviously critical that the authors carry this out convincingly to show that the rest of the analysis is robust, but the quality of the analysis and how it is presented can be improved.

RESPONSE: As suggested by the Reviewer, Figure 1 has been improved as described below.

Specific comments.

The comments on variability between sample origins are difficult to judge from Figure 1.

RESPONSE: As suggested, the revised Figure 1 is organized on the basis of sex so that readers can draw conclusions on sex-conserved gene expression across cell types. There were no notable differences in gene expression across datasets from various origins. Please note that the dataset is subjected to gene count normalization and cross origin batch correction.

Line 86 mentions "with TUID showing optimal sequencing metrics". This means that the HPAP did not show optimal sequencing metrics? This seems to be important and could be explained more clearly (eg in supplementary data).

RESPONSE: We thank the Reviewer for this important comment. We used the HPAP dataset as a metric to evaluate if our TUID sample dataset met current QC standards. We observed that our TUID samples were equal or superior in quality to HPAP with regard to median genes per cell, valid barcodes, reads mapped confidently to the genome, total genes detected/cell, median UMI counts/cell. This is shown in detail in supplemental Fig.1 and as suggested, this is now explained accurately in the revised manuscript (Line 90-91).

“As expected, sex chromosome-specific transcripts were expressed across male and female cell types” This is not newsworthy, a brief mention once or twice in the manuscript would suffice.

RESPONSE: As suggested by the Reviewer, we have made appropriate changes in the manuscript to reduce repetitions.

Most genetics research has come to the conclusion that the type of race definitions used by the authors are a poor way to describe human genetic diversity. Do the authors have DNA from the donors to improve their definitions?

RESPONSE: This study does not provide information on race definition or genetic diversity. In fact, the genotype data is absent for the HPAP dataset. We are defining race and ethnicity as defined in the U.S. by organ donors' self-identification, which is made clearer in the methods section. Self-identification of race (which does not necessarily make sense) includes Black, Asian or White, while self-identification of ethnicity is Hispanic or nonhispanic.

Line 97 This explanation can be improved, “for example, each β cell per donor was aggregated into one profile”. It is also unclear from the main text whether the authors are using usual definitions of 2pseudo-bulk" (apparently not the case).

RESPONSE: We are using the classical definition of pseudobulk profiling: The gene counts of each specific cell type per donor were aggregated into one gene count profile. This enabled us to control for the disproportionate cell numbers and related gene counts across donors. This also avoided the pseudo-replication of cells and their genes being repetitively sampled from a fixed donor. As suggested, this is made clearer in the revised manuscript (Lines 96-98) and is emphasized in the discussion of strengths (Lines 316-318).

Line 115. “canonical gene networks are conserved across endocrine and non-endocrine cell types independent of sex and race”. This is probably true, but this large figure does not really prove that.

RESPONSE: We agree, and as suggested, the revised Fig.1 focuses on side-by-side comparison of gene pathways in all cell types in males vs. females.

Figure 2 and text. The extended analysis of X and Y chromosome differences focuses on an extremely obvious fact that males and females differ in X and Y chromosomes

RESPONSE: We understand that this may seem obvious to the Reviewer, but the quantitative sex differences in X and Y chromosome gene expression and related chromatin accessibility has never been shown in islet cells (it is actually unknown in most cells). Further, the observation that some X chromosome genes escape inactivation in female islet cells and are

therefore overexpressed (from both X) is new and open novel research avenues for sex-based research. That some of these genes are chromatin remodelers that are involved in human disease like the histone demethylase KDM6A and are more accessible and expressed in female β and α cells is totally new. Similarly, that KDM5D, which is only expressed from the male Y chromosome, is overexpressed in male β and α cells (which seems obvious) is also new and its functional significance is unknown. Thus, these findings open avenues for sex-based research and precision medicine.

Figure 2. A major problem with claiming sex and race differences is the small number of samples and the inability to correct for confounders. Black male β cells showed higher cytokine signaling, and the authors state “suggesting black male β cells may exhibit a higher inflammatory response” But is this because a few black individuals have a specific disease association that lead to this pattern? (Or any other unknown variable).

RESPONSE: Despite our sample size, this is the largest orthogonal study of sex differences in human islets ever performed. Our statistical differential gene expression model corrects for confounders such as location of sequencing/10x library. Also, as explained above, we have used pseudo-bulking to increase stringency and eliminate the confounding effect of disproportionate cell numbers across donors. Thus, this is a stringent and appropriately powered study in the context of sex differences, and the largest study of that kind to date. Also, that a few black individuals exhibit this phenotype is highly relevant to precision medicine, since, as explained in the introduction, ketosis-prone diabetes is a form of T2D that occurs in a subset of male black individuals with severe insulin deficiency in a context of inflammation. As suggested by the Reviewer, this is emphasized in the discussion (Lines 305-308).

Figure 4. We found that Y-linked genes (SRY, RPS4Y1, UTY, TTTT14) in males and X-linked genes (KDM6A, XIST, DHRSX) in females were proximal to sex-associated cCREs. These descriptions again refer to something that is very obvious, how can females show accessibility at Y genomic regions?

This is true throughout the manuscript.

RESPONSE: Sex-specific gene expression, even if restricted to sex chromosomes is an important and novel finding with regard to transcriptional regulation across islet cell types. The question is why females show greater accessibility of some X chromosome regions when one X chromosome is supposed to be evolutionarily silenced in females to avoid bi-allelic X gene expression compared to males? Our data suggest that there are chromatin-associated mechanisms leading to escape of some X-linked gene inactivation in female islet cells. These X-linked genes showing higher expression in females are indeed genes that escape X-inactivation, which is expected to doubling gene dosage in females and have profound effects on the transcriptome. This is a sex-specific evolutionary mechanism that is essential to study in the context of biology and disease. In fact, as discussed, evidence demonstrates that these genes code for chromatin remodelers that play important sex-specific roles that influence a cell responsiveness to sex steroid hormone receptors or autonomous gene expression. In addition, that females exhibit a five to tenfold greater number of accessible binding sites for β and α cell transcription factors compared to males is a novel and unexpected finding.

Figure 5. The glucose/IBMX insulin secretion differences for black males are not convincing. In general all results in this figure would need more individuals, as well as the ability to consider confounders such as age, cause of death, disease, etc to be noteworthy

RESPONSE: Why isn't the data convincing if it is significant with the donors studied? This is the largest orthogonal assessment of human islets where the same donors were used for sc-RNA-seq, sc-ATAC-seq, dynamic insulin and glucagon secretion by perfusion using multiple secretagogues and bioenergetics analysis. As discussed above, the glucose/IBMX insulin secretion differences for black males is consistent with their propensity to develop β cell failure compared to white males.

The observation that females/males show a different T2D expression pattern is potentially interesting. The authors need to show that they are looking at sex, and not: batch, age, duration of diabetes, etc, etc. It would also be of interest to validate their findings in different datasets, with different sets of cases and controls, separately for accessibility and RNA. This is meant to a main section of the manuscript, but it is very superficial.

RESPONSE: We agree that this is a highly interesting finding. As highlighted above, we have included confounders in the design formula of our DEG test. We agree that it would be of interest to validate our findings in different datasets, with different sets of cases and controls, separately for chromatin accessibility and RNA. As the Reviewer knows, this would require another half a million \$ and 5 years of work. There is currently no other dataset of that kind available to compare male and female islets cells by scRNA-seq between nondiabetic and T2D subjects. Again, this is the largest orthogonal assessment of human islets where the same donors were used for sc-RNA-seq, sc-ATAC-seq, dynamic insulin and glucagon secretion by perfusion using multiple secretagogues and bioenergetics analysis.

In general the number of individuals from each class is missing in the figures/legends throughout the manuscript

RESPONSE: We have updated the demographics i.e. the number of individuals across each sex in the all of the figure legends.

The LD score regression analysis of male and female GWAS is interesting, and raises some questions:

- All cell types show similar enrichment. To better understand the significance of enrichments, it would be useful to show well matched data from other cell types that do not show enrichment.
- For the overlaps in individual loci, the same considerations discussed above are applicable. Are these sex-specific differences, or are they due to differences in duration, BMI, age, disease, glycemia, etc? How reproducible are these differences within this set of samples, and in an independent set?

RESPONSE: We have performed genomic enrichment analyses of our snATAC-seq open chromatin regions from well-matched data of all the 11 cell types present in the islets studied in this manuscript (Figure 7A). For the GWAS data to be well-matched as the Reviewer suggests, we would have to sequence samples of snATAC-seq open chromatin regions from other tissues orthogonally. There is no such well-matched data available.

Similarly, we have studied overlaps between open chromatin regions and loci which could contain multiple credible candidate genes associated with T2D. The analysis accounts for age and BMI as confounding factors.

It does not seem that the genetic differences are acting on different beta cell features. Are there any links between the sex specific differences in gene expression in beta cells and the sex specific genetic observations?

RESPONSE: Assuming that the Reviewer is referring to the GWAS study in Fig.6, there are no differences in β cells between males and females, but rather sex differences exists in other endocrine cells such as delta cells, which may influence the pathogenesis of T2D in a sex specific manner as discussed in the paper (Lines 250-252 and 309-316). We have performed a cell type specific, sex stratified, GWAS restricted to differentially accessible peaks across 3 metabolic traits: T2D, fasting glucose, and fasting insulin. This data is presented in Extended Data Fig.7 showing cell type-specific GWAS using DIAMANTE dataset with heatmaps of cell type or trait scaled enrichment metrics across sex. We found male and female β cells to be associated significantly with T2D risk while only female, not male β cells, were associated with elevated fasting glucose. In fact, while all islet endocrine cell types showed significant genomic enrichment in both male and female T2D GWAS, islet endocrine cells showed significant genomic enrichment in female only for fasting glucose GWAS. This unique data again suggests differences in the pathogenesis of hyperglycemia between males and females. We have included these results and a description of them in the revised manuscript (Lines 250-252 and 316-318).

Reviewer #2 (Remarks to the Author):

This study is an integrated investigation of the differences in male and female islet gene expression, open chromatin, GWAS sites, and islet function. The information to be gleaned from this study is important for understanding the sex differences in percentages of diabetes incidence and outcomes with late-onset diabetes. A few issues in presentation and interpretation should be addressed.

RESPONSE: We thank the reviewer for highlighting the important work performed in this manuscript.

1. The majority of the figures are too small to read at full-print size. When blowing them up, the version I had as a reviewer became pixelated. For example, in Figure 7A, it is extremely difficult to see the asterisks denoting significance.

RESPONSE: We have made formatting changes to make all of the figures readable in HD.

2. HPAP data was used and the grants supporting HPAP should be acknowledged. It is part of the terms of service for using HPAP data.

RESPONSE: We have included HPAP grant information in the acknowledgments.

3. Authors should discuss the limitations of ATACseq in examining open enhancers, especially since distal enhancers are essential for beta cell function and many T2D GWAS are located in those enhancers.

RESPONSE: As suggested by the Reviewer, we have developed the limitations section to discuss our restriction to proximal enhancers (Lines 327-329).

4. Line 70. While the study cited concluded that a minority percentage of the genes affected had sex hormone canonical binding sites, the authors' statement is too strong. Several studies have demonstrated chromatin binding of ER-alpha without an ERE, especially for cell-specific binding sites.

RESPONSE: We have rephrased this part of the introduction.

5. For Extended Figure 3, in "e", the legend says "aggregated read coverage", but the figure says "gene activity". For "f" the legend says "aggregate gene activity" and the Figure says "chromatin accessibility". This is confusing. In addition, how is "aggregate gene activity" defined?

RESPONSE: Aggregate gene activity is the pseudobulked gene accessibility profiles across all cells in a particular cell type for each donor. We have used the term chromatin accessibility density profile for simplicity. In the revised Figure 3 legend we explain this in greater detail.

6. Error bars are not defined in all figure legends.

RESPONSE: All error bars are now defined.

7. The abstract should be re-tooled for clarity. The second and third sentences are indirect and small grammar errors further complicate understanding.

RESPONSE: The abstract has been reworded for clarity and consistency.

Reviewer #2 (Remarks on code availability):

I am not a programmer.

Reviewer #3 (Remarks to the Author):

The manuscript reports the collection and analysis of single-cell RNA sequencing datasets from human pancreatic islets of donors, categorized by gender (male and female) and race (white, black, and Hispanic). The study aims to interrogate transcriptional and chromatin accessibility profiles by identifying genes conserved or upregulated by sex, race, and type 2 diabetes. This study provided single cell datasets from 15 healthy donors and incorporated additional datasets from the HPAP database. The analyses include an in-depth characterization of human islets, highlighting gender- and endocrine-related factors upregulated in each condition. Specifically, genes linked to male or female gender are enriched in their respective conditions, as expected. While the manuscript offers a detailed characterization of these samples, it lacks significant biological insights from the analyses, questioning the study's novelty. The findings were not validated experimentally, and the wet lab experiments did not yield results relevant to the field. In terms of resource utility, the study did not present new findings that contribute to the field's development, and the sequencing performed does not introduce new avenues to the existing database. Overall, I cannot recommend this manuscript for publication in *Nature Metabolism*.

RESPONSE: First, this study does not include “gender” but biological “sex” as sex chromosomes are studied. We disagree with the assessment that “genes linked to male or female gender are enriched in their respective conditions, as expected”. It cannot be expected that X-linked genes are enriched in females more than males since one X chromosome is silenced to avoid biallelic expression, and X-escape genes (expressed from both alleles) vary from cell types to cell types. Therefore, the sex-specific sex chromosome gene expression and their related chromatin accessibility in islet cells is new. Further, that females exhibit a five to tenfold greater number of accessible binding sites for β and α cell transcription factors compared to males is a novel and unexpected finding.

The comment that “In terms of resource utility, the study did not present new findings that contribute to the field's development” is not true and unfair as this study provides a comprehensive resource available as an online tool hosted on the metabolic genome atlas (CMDGA), which is the largest collection of high throughput metabolic data to date. Researchers can rapidly and freely look for gene expression and accessible chromatin in all cells in a sex-specific manner, generate graphics and download them as high-resolution images for publication or presentation. It is a unique resource for the community that has never been generated before. No such dataset analysis tool exists which can look at single cell/nuclei data in islets in a sex-specific manner. How can this resource not present new findings that contribute to the field's development?

Major

1. Figure 1E – The manuscript does not clearly convey the purpose of this figure. Although the figure caption provides an explanation, it is difficult to follow within the main text.

RESPONSE: As suggested, we have revised the wording in the manuscript to describe Fig.1E more appropriately. This figure denotes the number of all upregulated genes for each specific cell type in comparison to other cell types.

2. Figure 1G – It is unclear how the analysis depicts differences or similarities between sex and race in key endocrine cell populations. For example, line 106 suggests that certain gene sets are enriched independently of sex and race. However, the analysis does not highlight this aspect but rather shows that the expected gene sets are present in the identified population. Please expand on the implication and significance of this analysis and its meaning for the study.

RESPONSE: As suggested, the revised Fig.1 now includes a side-by-side comparison of gene pathways in all cell types in males vs. females. There are conserved gene expression programs irrespective of sex in each cell type. The gene expression programs translate to gene pathways which are also conserved irrespective of sex. This is now depicted clearly in Fig. 1G and F.

3. Figure 2C and 2D – What are the other motivations for this analysis apart from the differences in the expression of sex-related genes? Do the other genes identified have implications for beta/alpha cell identity, function, or diabetes?

RESPONSE: For each cell type, the left panel shows that DEG mostly involve X-linked genes in terms of fold-change and p-value. Thus, we have created the right panel to show sex differences in autosomal DEGs. The few sex-associated autosomal genes in α/β cells of non-diabetic donors exhibited a small effect size compared to sex-associated X and Y genes (Fig. 2c and d), which as suggested is discussed in the revised manuscript (Line 139-140).

4. Figure 2E – Some of these gene sets seem arbitrary. Please include gene sets or pathways relating to pancreatic islets and/or diabetes. For example, do sex or race have any relevant differences in pathways that contribute to beta cell insulin secretion, metabolism, maturation, or health?

RESPONSE: The gene sets shown in Fig. 2E are not arbitrary. The panel is a classical way to represent all the significant pathways activated assessed by gene ontology, most of which are sex-linked. These pathways have a direct impact on beta/alpha cell function.

5. Figure 3A-F – This type of analysis is not novel, as it has been performed previously by the same group. These figures should focus on the chromatin accessibility of key genes, comparing sex and race.

RESPONSE: We disagree. This analysis is novel and none of us has ever performed it before. Fig.3A-F are necessary to show the appropriate integration and cell type classification in snATAC-seq cells. It establishes the basis for asking the question, which the reviewer alludes to, a focus of F4 and F3g-j.

6. Figure 3 – The paper initially focused on racial differences, including white, black, and Hispanic populations. However, this figure only covers two races.

RESPONSE: Our snATACseq data of 15 donors has only 2 races, Black and White because of the paucity of non-White organ donors in the US (See answer 7 below). We focused on Black and White because of Black individuals have a propensity to beta cell failure (Ketosis-Prone diabetes) compared to Whites.

7. To appropriately claim that this study examines racial aspects, the Asian population should be included.

RESPONSE: Too few organ donors are none white in the US, and when they are they are mostly Black. Asian individuals have the lowest organ donation registration rates in the US (PMID: 30968472). In the HPAP samples sequenced, no Asian dataset was available or met the criteria for inclusion based on our selected BMI and age. Today only 2 Asian donors are available in the HPAP. This limitation is beyond our control. We decided to focus on Black vs White because of the implication for KPD as discussed in the introduction.

8. The findings in Figure 3 are similar to previous publications, including Wang et al. (Nature Genetics).

RESPONSE: The manuscript highlighted by the Reviewer (PMID: 37231096) does not study sex differences in accessible chromatin regions and includes a completely different dataset. The current study 1) utilizes a novel snATACseq+scRNAseq dataset, which has not been analyzed before and 2) expands on previous work to analyze sex differences. As described above, Fig. 3 is necessary to establish QC for later investigation and to show that our dataset is appropriate for the study of sex-specific differences.

9. Figure 4 – Please expand on the significant implications of the reported genes and TF-specific motif accessibility.

RESPONSE: We have expanded on the implications in metabolic disease for the reported TF-specific motifs in the revised discussion.

10. Figure 5 – What kind of replicates are these? Are the measurements represented by multiple donors?

RESPONSE: We performed hormone secretion and bioenergetics measurements in the same TUID donors than those that have been sequenced and studied in Fig.1 to 4 to maintain orthogonality in the study design. Each dot is a separate biological donor.

11. Figure 5B, D, F, H – Static glucose-stimulated insulin secretion should be performed to obtain an absolute measurement of insulin secretion rather than calculating the AUC from dynamic secretion.

RESPONSE: We disagree, the measurement of dynamic insulin secretion by perfusion is the gold-standard for all islets biologists for assessment of islet hormone secretion *in vitro*. Insulin secretion from pancreatic beta-cells is bi-phasic and this feature can only be recapitulated during perfusion. Further, the static incubation of islets in their hormone secretion is not physiological as secreted hormones are normally washed out by blood flow, which is recapitulated only by perfusion. We use a state-of-the art microfluidic dynamic perfusion platform which enables the study of islet cell secretion dynamics at high temporal resolution, and under varying conditions of insulin and glucagon secretagogues. Our highly consistent results are in line with our previous reports (PMID: 37200193, PMID: 32601271) as well as

previous reports from leaders in the field of islet biology (PMID: 31632354, PMID: 35642629). In addition perfusion is used to evaluate secretion profiles of islets utilized for transplantation, representing the highest level of stringency possible (PMID: 18351020, PMID: 25648831). Representing hormone secretion as a function of AUC is appropriate as it represents the accurate secretion index in response to a secretagogue (PMID: 33737253).

12. Figure 5 should also include functional assessments from other races.

RESPONSE: Please see response to Questions 7 and 10.

13. Figure 6 and 7 – Some of these genes should be validated with wet lab experiments.

14. The sample size of this study is small, making it difficult to fully encompass the racial diversity aspect. Other major races are missing, including Hispanics and Asians.

15. The study lacks functional and wet lab validation of pathways identified when comparing islets of different sexes and races.

16. The analysis lacks an in depth mechanistic study investigating the gender and race related factors in islet biology.

RESPONSE to 13-16: Our goal was to study large sequencing datasets to provide a general regulatory architecture of sex differences in human islet cells. We included black individuals thus increasing genetic diversity of the study. It establishes an important resource for future discovery by the scientific community. The functional “Wet lab” validation of all pathways discovered in this study and the in-depth mechanistic study investigating the gender and race related factors in islet biology is out of the scope of the current manuscript as it would be the topic of several studies and manuscripts. In fact, it is the ongoing mission of the NIH via the Human Pancreas Analysis Program (HPAP) to perform large orthogonal studies similar to those performed in this manuscript. We are the first to study sex differences at this scale utilizing multiple genomic and metabolic assays in islets. This study encompasses over a decade of meticulous sample collection, ensuring the inclusion of black individuals thus increasing genetic diversity and the analysis is completely orthogonal in nature.

Minor

1. Line 88 - Figure 1C – In the text, the claim that there is an even distribution in cellular clusters could be made more robust using quantitative measures such as bar plots or pie charts rather than qualitatively in a UMAP figure. Can you provide this information as well? (This can also be included in the supplement.)

RESPONSE: This information is already provided. We have quantitative segmented barplots for celltype cluster distribution across each donor based on cell number or proportion (F1d, Extended data F1b, and F3c).

2. Figure 1E - The figure shows conserved DEGs upregulated in each cell population. This is not explicitly stated in the main text, making it difficult to follow.

RESPONSE: We have updated the manuscript to state this in the text.

3. Provide abbreviation details of all abbreviations used in te figure captions.

RESPONSE: Abbreviations have been added to all figure legends where appropriate.

4. Missing missing color legend for the dataset type and category for data type in multiple figures.

RESPONSE: A description of color legends has been added to figure legends.

5. Ensure clarity in the main text.

RESPONSE: The text has been refined for clarity.

Reviewer #3 (Remarks on code availability):

The codes deposited are appropriate.

Dear Franck,

Thank you for submitting your revised manuscript to The EMBO Journal. Your manuscript has now been seen by one of the original reviewers, who now finds that their previous concerns have been sufficiently addressed and recommends publication after toning down of the statements regarding race-related differences.

I will therefore be happy to accept the manuscript for publication in The EMBO Journal after its minor textual revision as requested by the reviewer and reformatting along the guidelines below and in the attached document.

Please feel free to contact me if you have any further questions regarding this final revision. Please use the link below to upload the revised files.

Thank you for the opportunity to consider your work for publication, and I look forward to receiving your revised manuscript.

With best regards,

Ieva

- a point-by-point response to the referees' comments, with a detailed description of the changes made (as a word file).

- a word file of the manuscript text.

- individual production quality figure files (one file per figure)

- a complete author checklist, which you can download from our author guidelines

(<https://www.embopress.org/page/journal/14602075/authorguide>).

- Expanded View files (replacing Supplementary Information)

- a Reagents and Tools Table as part of the Methods section, which can be downloaded from our author guidelines

(<https://www.embopress.org/page/journal/14602075/authorguide#structuredmethods>)

We realize that it is difficult to revise to a specific deadline. In the interest of protecting the conceptual advance provided by the work, we recommend a revision within 3 months (9th Jan 2025). Please discuss the revision progress ahead of this time with the editor if you require more time to complete the revisions.

Referee #1:

The authors have addressed some of the reviewers' suggestions and improved the manuscript. I still have many of the same comments I pointed out in the original review, including the fact that this is an enticing study which points to sex-specific differences in the islet pathophysiology and genetic susceptibility of T2D. I believe it will be of interest to EMBO J readers.

In contrast to the analysis of sex differences, many aspects of the race comparisons are weaker. The authors still refer to "hispanic" as a "race" which is pretty groundless, regardless of how the US census feels about this. Furthermore, with few donors the conclusions on differences between black vs. white donors are tenuous. For example, the slight differences in insulin secretion seem based on three black individuals who may differ from white individuals in who knows how many ways apart from their skin pigmentation (cause of death, age, medication...). Insulin secretion is highly variable between donors, but this is also applicable to other types of comparisons. The figure legends I checked don't seem to state how many individuals in each race group. Since the authors feel strongly about keeping these types of comparisons, they might want to tone down or reword any references to them.

The authors addressed the remaining issues.

Dear Franck,

Thank you for submitting a revised version of your manuscript. I am afraid that there remain a few formatting aspects as outlined below that still need to be sorted out in the manuscript before I can formally accept it for publication:

1. Please reduce the number of keywords to five.
2. Please check that the funding information is correct and identical both in the manuscript and our online system. Currently, P20GM152305, Tulane Center of Excellence in Sex-Based Biology & Medicine are missing in our online system.
3. On the front page, please check the numbering of author affiliations - currently number "9" is missing.
4. Please remove the "Materials availability" section.
5. Please move "Methods" after "Discussion".
6. In the Data Availability section, please add resolvable links to the datasets. More information about the format of this section can be found here: <https://www.embopress.org/page/journal/14602075/authorguide#dataavailability>.
7. CRedit has replaced the traditional author contributions section because it offers a systematic, machine-readable author contributions format that allows for more effective research assessment. Please remove the Authors Contributions from the manuscript and use the free text boxes beneath each contributing author's name in our online submission system to add specific details on the author's contribution. More information is available in our guide to authors.
8. Please rename "Declaration of interests" section into "Disclosure and competing interests statement" (further info: <https://www.embopress.org/page/journal/14602075/authorguide#conflictsofinterest>).
9. Please remove Reagents and Tools Table from the manuscript text and upload as a separate .docx file choosing the file type "Reagent Table".
10. In the Appendix file, please provide page numbers in the table of contents.
11. Please provide a short "synopsis" of your study consisting of a brief (1-2 sentences) summary of the findings and their significance and 3-4 bullet points highlighting key results.
12. Our data editors have flagged the following issues in figure legends that need correcting:
 - Please provide the exact p values in the legends of figures 5b, j, l; 7a.
 - Please indicate the statistical test used for data analysis in the legends of figures 1f; 2c-f; 3h; 4d, f, h; 5b, j, l; 6b-c, e, g; 7c.
 - Please provide information on the number and nature (e.g., biological or technical) of replicates in the legends of figures 2c-d; 4f; 6b-c.

Please feel free to contact me if have any questions regarding this final revision. Thank you again for giving us the chance to consider your manuscript for The EMBO Journal. I look forward to receiving the revised version.

With best wishes,

Ieva

Ieva Gailite, PhD
Senior Scientific Editor
The EMBO Journal
Meyerhofstrasse 1
D-69117 Heidelberg
Tel: +4962218891309

We realize that it is difficult to revise to a specific deadline. In the interest of protecting the conceptual advance provided by the work, we recommend a revision within 3 months (21st Jan 2025). Please discuss the revision progress ahead of this time with the editor if you require more time to complete the revisions.

The authors addressed the remaining editorial issues.

Dear Franck,

Thank you for addressing the final editorial points. I am now pleased to inform you that your manuscript has been accepted for publication.

Before we forward your manuscript to the publishers, I would like to suggest minor edits in the manuscript title, abstract and synopsis. I have also written a blurb that will accompany the title of your manuscript on our website. Please take a look at the text below and in the attached file and let me know if any further edits or corrections are needed.

Alternative title:

Sex-specific differences in the regulatory architecture of pancreatic islets from type 2 diabetic patients

Blurb:

Single cell-level analyses of human islet cell transcriptome and chromatin accessibility reveal sex-specific differences in β cell gene regulation in healthy and diabetic donors.

Synopsis

Diabetic patients show sex-specific differences in insulin secretion; however, the underlying mechanisms remain unclear. This study analyses the single cell regulatory architecture of human pancreatic islets and establishes biological sex as a genetic modifier of β cell function and the pathogenesis of type 2 diabetes (T2D).

- Islet cells from nondiabetic (ND) donors exhibit sex differences in chromatin accessibility and gene expression, predominantly affecting sex chromosomes.
- Islets from T2D donors not only exhibit similar differences in sex chromosomes as ND donors, but also show sex-specific differences in autosomal gene regulation.
- Comparing β cells from T2D and ND donors, gene enrichment suggests a role for mitochondrial failure in the pathogenesis of T2D in females.
- A GWAS restricted to differentially accessible chromatin regions across T2D traits identified regions that overlap with T2D-associated variants in a sex- and cell type-specific manner.

If you have any questions, please do not hesitate to contact the Editorial Office. Thank you for this contribution to The EMBO Journal and congratulations on a nice study!

Best wishes,

leva

leva Gailite, PhD
Senior Scientific Editor
The EMBO Journal
Meyerohofstrasse 1
D-69117 Heidelberg
Tel: +4962218891309
i.gailite@embojournal.org

>>> Please note that it is The EMBO Journal policy for the transcript of the editorial process (containing referee reports and your

response letter) to be published as an online supplement to each paper. If you do NOT want this, you will need to inform the Editorial Office via email immediately. More information is available here: https://www.embopress.org/transparent-process#Review_Process